# White matter disconnection of left multiple demand network is associated with post-lesion deficits in cognitive control

Jiefeng Jiang [1,2,3] ✉, Joel Bruss[4,5], Woo-Tek Lee[1,2,6], Daniel Tranel[1,3,4] & Aaron D. Boes [3,4,5,7] ✉

Cognitive control modulates other cognitive functions to achieve internal goals and is important for adaptive behavior. Cognitive control is enabled by the neural computations distributed over cortical and subcortical areas. However, due to technical challenges in recording neural activity from the white matter, little is known about the anatomy of white matter tracts that coordinate the distributed neural computations that support cognitive control. Here, we leverage a large sample of human patients with focal brain lesions (n = 643) and investigate how lesion location and connectivity profiles account for variance in cognitive control performance. We find that lesions in white matter connecting left frontoparietal regions of the multiple demand network reliably predict deficits in cognitive control performance. These findings advance our understanding of the white matter correlates of cognitive control and provide an approach for incorporating network disconnection to predict deficits following lesions.

Cognitive control refers to a set of cognitive functions that align neural processing with internal goals[1,2]. Cognitive control is crucial in every-day behaviors such as switching between tasks[3], suppressing habitual but goal-irrelevant responses[4], and canceling potent or initiated actions[5]. There has been extensive cognitive neuroscience research investigating the grey matter (GM) substrates of cognitive control[6,7]. It has been shown that cognitive control is supported by distributed cortical and subcortical areas[8,9] with key cortical regions often grouped together as a brain network (e.g., the 'multiple demand' net-work MDN[8,10,11]). Lesions in these areas are followed by deficits in cognitive control[12,13]. A more complete understanding of cognitive control and its role in adaptive behavior requires that we understand not only these GM regions that have been extensively investigated but also the white matter (WM) tracts that connect them to coordinate neural computations in the distributed GM regions. However, due to the technical difficulties in recording neural activity from the WM,

there has been limited empirical evidence of the WM substrates underlying cognitive control. Although WM structural measures are correlated with cognitive control performance[14–17], evidence showing a causal role of these WM regions in cognitive control is lacking and requires further investigation.

In this study, we investigate the WM correlates of cognitive control in a large sample of 643 individuals with focal brain lesions that underwent neuropsychological testing of tasks requiring cognitive control. In light of the aforementioned research on cognitive control in the GM and recent studies showing that WM disconnection outperforms GM measures in predicting post-lesion behavioral outcomes[18], we hypothesize that WM disconnection of tracts connecting the cognitive control network will be associated with impaired cognitive control performance. Here, we are interested in domain-general cognitive control, or the cognitive control processes shared among different tasks[19–21]. To measure domain-general cognitive control in

[1]Department of Psychological and Brain Sciences, University of Iowa, Iowa City, IA 52242, USA. [2]Cognitive Control Collaborative, University of Iowa, Iowa City, IA 52242, USA. [3]Iowa Neuroscience Institute, University of Iowa, Iowa City, IA 52242, USA. [4]Department of Neurology (Division of Neuropsychology and Cognitive Neuroscience), Carver College of Medicine, Iowa City, IA 52242, USA. [5]Department of Psychiatry, Carver College of Medicine, Iowa City, IA 52242, USA. [6]Behavioral-biomedical Interface Training Program, University of Iowa, Iowa City, IA 52242, USA. [7]Department of Pediatrics, Carver College of Medicine, Iowa City, IA 52242, USA. ✉e-mail: jiefeng-jiang@uiowa.edu; aaron-boes@uiowa.edu

behavior, we used two neuropsychological tests: the trail-making test (TMT) that requires flexible switching between two tasks (i.e., linking letters/numbers in ascending order), and the Stroop task that requires inhibition of a habitual response (i.e., reading a word) and strengthening a novel but task-relevant response (i.e., naming the ink color of a word). We used a cognitive control network in the hypothesis as an a priori network of interest. Out of the many networks involved in cognitive control, we used the MDN[8] as an operationalization of the cognitive control network. The MDN is defined based on multiple tasks that rely on different aspects of cognitive control. The tasks defining the MDN are also different from the neuropsychological tests used in this study, thus further ensuring the test of domain-general cognitive control.

To test our hypothesis, we performed both data-driven and hypothesis-driven analyses. In the data-driven analyses, we employed a machine learning approach to identify patterns of cerebral cortex disconnection that maximally predict cognitive control performance. In the hypothesis-driven analyses, we used the same methods as in the data-driven analyses but limited the analyses to WM tracts of the MDN, in accordance with the hypothesized importance of the WM tracts connecting the MDN in cognitive control. We then tested the hypothesis by evaluating the predictive performance of the hypothesis-driven analyses relative to the data-driven analyses (i.e., if the hypothesized MDN regions explain the same amount of variance in cognitive control performance as the data-driven analyses spanning the whole cerebral cortex, this would support the notion that MDN is critical in cognitive control). In the analyses, the machine learning approach first learned the predictive model from the training data. The model was then subjected to out-of-sample, within- and between-task cross-validation analyses to ensure external validity and generalizability. To preview the results, we find that lesions involving WM tracts connecting left frontoparietal regions of the MDN are reliably associated with cognitive control impairment. Moreover, cognitive control performance is better predicted using disconnection in WM anatomical connectivity compared to disconnection in GM functional connectivity, and by left hemisphere (LH) than right hemisphere (RH) lesions.

## Results

To investigate the lesion anatomy and disconnection patterns predictive of impaired cognitive control, we acquired neuropsychological test data and imaging scans from three samples (total $n = 643$ unique subjects, Table 1) of patients with acquired focal brain lesions. Lesions of multiple etiologies were included and are summarized in Supplementary Table 1. Behavioral performance of cognitive control was measured using the TMT and Stroop tasks (see Supplementary Note 1 for tests of construct validity). To better capture cognitive control, analyses were conducted on test scores

that control for general processing performance (i.e., TMT: part B score minus part A score; Stroop: interference score; see Methods for detail). Behavioral performance was worse in test conditions requiring more cognitive control (Supplementary Note 2) and was not correlated with common co-occurring symptoms (Supplementary Note 3). To ensure the external validity of the findings, we split the sample into three groups (two groups with TMT scores and one with Stroop scores) prior to conducting analyses. This allowed for three analyses with increasing levels of generalization (Fig. 1a): within-sample (group 1), cross-sample (trained on group 1 and tested on group 2) and cross-task validation (trained on TMT on group 1 and tested on Stroop performance in group 3; neuropsychological scores were standardized to allow for cross-task prediction). Although some subjects with TMT data analyzed in group 2 were also included in group 3 with Stroop scores, all analyses ensured that the training and test data used data from different participants. The duration between lesion onset and behavioral testing was not correlated with cognitive control performance (see below) in any of the three samples (TMT-1: $r = -0.007$, $p > 0.89$; TMT-2: $r = 0.054$, $p > 0.36$; Stroop: $r = -0.079$, $p > 0.23$), indicating that it is unlikely to be a confounding factor in predicting cognitive control performance. The lesion distribution of each participant group is shown in Table 1 and Fig. 1b.

### Left prefrontal lesions correlate with cognitive control deficits

Prior to the disconnection analysis, we first identified lesion locations associated with impaired cognitive control performance. Specifically, individual structural brain scans were manually traced to delineate the location of the lesions. The lesion locations were then transformed to a template brain for cross-subject comparisons. Using these lesion locations as masks and behavioral data, we performed lesion-symptom mapping with multivariate sparse canonical correlations (LESYMAP[22]) to identify lesion locations that predict behavioral scores (Fig. 2a). This analysis demonstrated that cross-subject variance in the TMT scores (combining TMT-1 and TMT-2 samples) can be most reliably explained by lesions of the left frontoparietal WM, with additional findings in bilateral frontal, parietal, and posterior temporal areas ($r = 0.27$, $p = 5 \times 10^{-12}$; Fig. 2b). In comparison, lesions that involve the right and left prefrontal cortex were associated with impaired Stroop scores ($r = 0.30$, $p = 4 \times 10^{-6}$, Fig. 2b, in red). Importantly, the supero-posterior aspect of the left inferior frontal gyrus (area IFSp according to the HCP atlas[23]) and the underlying WM of the frontal aslant tract is a region included in both the TMT and Stroop lesion-symptom maps and a latent variable derived from both assessments. As such, this region, when lesioned, is associated with greater impairments in both TMT and Stroop performance (Fig. 2b, in green, center MNI coordinates = −44, 17, 22). To test if the amount of overlap between the TMT and Stroop lesion-symptom map is significantly above chance, we constructed a null distribution by randomly permuting the locations of clusters that survived multiple comparisons for each lesion-symptom map within the brain mask and computed the amount of overlap over 1000 iterations. The observed overlap is significantly larger than the median of the null distribution ($p = 0.005$), suggesting that the shared area is unlikely to be a coincidence. Next, we evaluated the location of significant lesion-symptom map results relative to an a priori region of interest (ROI) representing the WM tracts connecting the MDN (Fig. 2c). To this end, we first derived individual streamline maps representing WM connections from each cortical ROI of the MDN using deterministic tractography in a normative diffusion MRI dataset. The individual maps were combined using principal component analysis (PCA) to identify common regions of WM connectivity of the MDN ($n = 209$, Fig. 2d). We found that the lesion-symptom maps overlap with the WM connectivity of the MDN (Supplementary Note 4). In the following analyses, we tested the hypothesis using measures of disconnection.

## Table 1 | Demographic information for each of the three samples

| Sample | Group 1 (TMT-1) | Group 2 (TMT-2) | Group 3 (Stroop) |
|---|---|---|---|
| N | 335 | 287 | 229 |
| Age (SD) | 54.54 (15.51) | 51.71 (14.38) | 51.09 (11.37) |
| Sex | 149(F)/186(M) | 139(F)/148(M) | 117(F)/112(M) |
| Education, years (SD) | 13.46 (2.82) | 13.83 (2.69) | 13.86 (2.69) |
| Handedness | 300(R)/26(L)/9(A) | 256(R)/24(L)/7(A) | 207(R)/17(L)/5(A) |
| Median (SD) time from onset to scan, in months | 13.3 (63.2) | 10.1 (91.9) | 18.5 (98.7) |
| Lesion laterality | 115(R)/157(L)/63(B) | 110(R)/110(L)/67(B) | 85(R)/83(L)/61(B) |

*R* right-handed, *L* left-handed, *A* ambidextrous.

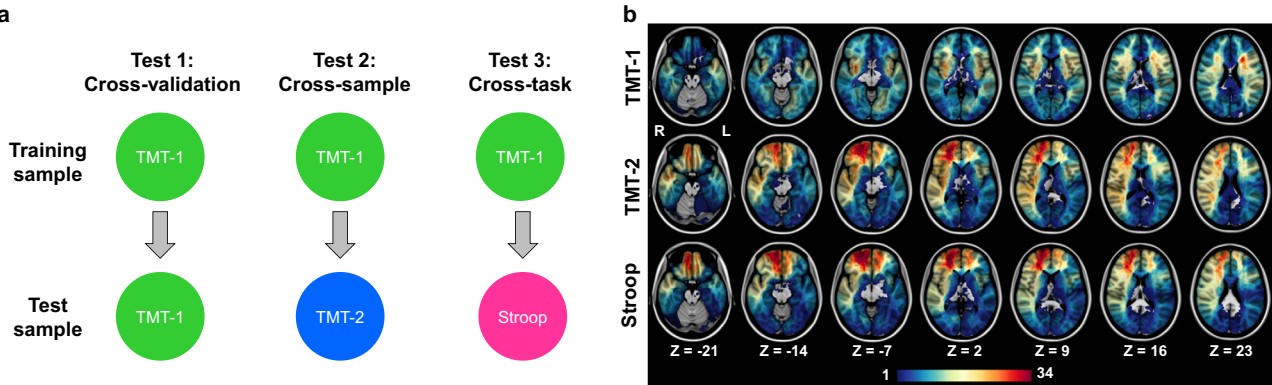

**Fig. 1 | Participant groups. a** The training and test samples used in each of the three analyses with increasing levels of generalization (i.e., within-sample cross-validation, cross-sample generalization and cross-task generalization).

**b** Visualization of lesion distributions in each group. The color scale indicates the number of patients having overlapping lesions. TMT trail-making task.

## Connectome-based predictive analysis overview

Next, we evaluated how disruption of brain connectivity contributes to cognitive control performance. In addition to the location of the WM used in lesion-symptom mapping, we further leveraged the WM connectivity to different cortical regions using AC maps seeded from different cortical ROIs (Fig. 2c). The added information helps uncover cognitive control-impairing disconnection patterns that lesion-symptom mapping may be unable to identify. To identify disconnection patterns that predict cognitive control performance, we adopted the connectome-based predictive modeling approach (CPM[24,25]). In order to adapt this CPM method for a lesion analysis, we first estimated the extent to which each individual lesion was connected to each cortical ROI (Fig. 3; see the "Methods" section for detail). To this end, each cortical ROI from the Human Connectome Project (HCP) atlas[23] was used as a seed region to generate an AC and a functional connectivity (FC) map using tractography and resting-state FC data. These maps were derived from an independent dataset of healthy adults. Next, each lesion mask was overlaid with the AC and FC maps derived from each ROI to produce an AC-based disconnection score (Fig. 3a) and an FC-based one. The disconnection score was the sum of the voxel-wise connectivity values on a brain-wide connectivity map (i.e., AC or FC) that intersected with the lesion mask. This procedure resulted in a score for each individual lesion reflecting either AC or FC disconnection for each of the 360 ROIs. With CPM, the ROIs whose lesion disconnection scores are significantly correlated with neuropsychological scores in the training data (Fig. 3b) are selected to predict neuropsychological scores in independent test data (Fig. 3c, d, see the "Methods" section for detail).

In our analyses, training and test data were always independent to avoid model overfitting. The performance of each prediction was assessed by comparing predicted and actual behavioral test data using Akaike Information Criterion (AIC), with a lower AIC indicating better prediction. All AICs are listed in Table 2. To assess the effect size, we calculated the ratio of prediction error between two predictive models (denoted as $R$, see the "Methods" section). For example, an $R$ score of 1.03 indicates that prediction error from the worse model is, on average, 1.03 times, or 3% larger than, that from the better model for each participant.

## AC predicts cognitive control deficits better than FC

We first compared CPM performance using AC and FC disconnection scores. Across all three analyses with increasing levels of generalization, AC disconnection scores consistently outperformed FC disconnection scores in predicting neuropsychological measures of cognitive control (cross-validation within TMT-1: $\Delta AIC = -18.6$, $p = 9 \times 10^{-5}$, $R = 1.03$; TMT-1 → TMT-2: $\Delta AIC = -16.2$, $p = 0.0003$, $R = 1.03$;

TMT-1 → Stroop: $\Delta AIC = -28.7$, $p = 6 \times 10^{-7}$, $R = 1.07$; Fig. 4a). Furthermore, adding FC disconnection scores to AC disconnection scores did not improve prediction performance, as there was no statistically significant difference in AICs between CPMs using AC disconnection scores only and CPMs using both AC and FC disconnection scores in any of the three analyses (all $p$s > 0.31; Fig. 4a). Therefore, we focused on AC disconnection scores in the next analyses.

## LH predicts cognitive control deficits better than RH

We further tested the lateralization of prediction performance and observed that AC disconnection scores in the LH were better predictors than those in the RH (cross-validation within TMT-1: $\Delta AIC = -14.7$, $p = 0.0007$, $R = 1.02$; TMT-1 → TMT-2: $\Delta AIC = -12.0$, $p = 0.0024$, $R = 1.02$; TMT-1 → Stroop: $\Delta AIC = -41.3$, $p = 1 \times 10^{-9}$, $R = 1.10$; Fig. 4b). Across the three analyses, prediction performance did not significantly improve by adding RH disconnection scores to LH disconnection scores (all $p$s > 0.50; comparison between both hemispheres and LH in Fig. 4b), suggesting that RH disconnection scores did not provide additional information for prediction beyond that provided by LH disconnection scores. The same patterns were observed when controlling for the lesion volumes between LH and RH across subjects (Supplementary Fig. 2). Similar to the findings above, LH AC disconnection scores consistently outperformed LH FC disconnection scores in predicting cognitive control performance (cross-validation within TMT-1: $\Delta AIC = -19.2$, $p = 7 \times 10^{-5}$, $R = 1.03$; TMT-1 → TMT-2: $\Delta AIC = -17.2$, $p = 0.0002$, $R = 1.03$; TMT-1 → Stroop: $\Delta AIC = -25.5$, $p = 3 \times 10^{-6}$, $R = 1.06$; Fig. 4b).

## MDN outperforms the whole brain in prediction performance

To test our hypothesis that variance in cognitive control performance was related to WM disconnection of the MDN, we constrained the CPM model to the subset of ROIs of the MDN (17% of cortical ROIs). If the MDN connectivity is critical to cognitive control, then prediction performance should be similar to that using all cortical ROIs. In contrast, if regions outside the MDN are critical to the predictive performance of the model, the model's performance will suffer when constrained to the MDN ROIs. When tested within the TMT, MDN displayed numerically worse prediction performance. However, this difference did not reach statistical significance (cross-validation within TMT-1: $\Delta AIC = 3.7$, $p > 0.13$, $R = 1.006$; TMT-1 → TMT-2: $\Delta AIC = 3.3$, $p > 0.15$, $R = 1.006$; Fig. 4c). Importantly, when tested between the TMT and the Stroop task, MDN yielded better prediction performance than all ROIs ($\Delta AIC = -12.2$, $p = 0.0022$, $R = 1.02$; Fig. 4c). When we expanded the comparison using CPM on both AC and FC disconnection scores, the MDN still performed similarly to or better than all ROIs (cross-validation within TMT-1: $\Delta AIC = 4.1$, $p > 0.11$, $R = 1.006$; TMT-1 → TMT-2:

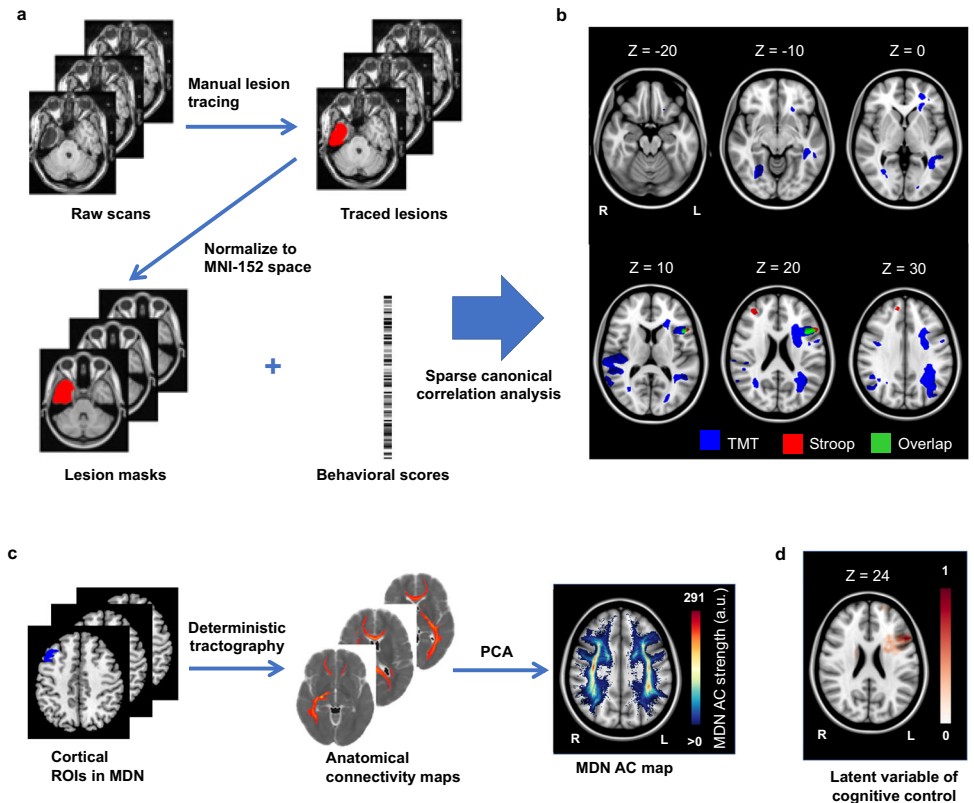

**Fig. 2 | Multivariate lesion-symptom mapping analysis. a** Steps of generating lesion masks and the lesion-symptom mapping (LESYMAP) analysis. **b** Multivariate lesion-symptom mapping results. **c** Procedure of generating region of interest (ROI)-wise anatomical connectivity (AC) connectivity maps and the multiple demand network (MDN) AC map. **d** Lesion-symptom map results using a latent variable of cognitive control. Color encodes voxel-level loading in predicting symptoms. TMT trail-making task.

ΔAIC = 4.0, $p > 0.11$, $R = 1.007$; TMT-1 → Stroop: ΔAIC = −12.0, $p = 0.0024$, $R = 1.03$; Fig. 4d). The better performance of MDN in cross-task generalization may be due to overfitting to the TMT task when all ROIs were used. That is, some non-MDN ROIs that were helpful only in predicting TMT scores were selected in the model and this led to reduced performance when tested on Stroop scores. As an assessment of the goodness of prediction, predictions from MDN were positively correlated with actual neuropsychological measures in the cross-sample and cross-task generalization tests (TMT-1 → TMT-2: $r = 0.26$, $p = 9 \times 10^{-6}$; TMT-1 →;Stroop: $r = 0.21$, $p = 0.0015$; Fig. 4e).

### Evaluation of specificity

In order to evaluate the specificity of this analysis for the MDN, we performed additional tests. First, we conducted a control CPM analysis using a set of randomly selected ROIs such that the new ROI set and the MDN have the same number of ROIs. The AIC for the random ROI set was then recorded. The procedure was repeated for 1000 times, producing an approximation of the null distribution of AIC. MDN significantly outperformed the median of the null distribution ($p = 0.001$, Fig. 4f). Second, we compared predictions from the CPM using MDN to other variables less specific to cognitive control. This includes predictions derived from the edge density map[18], which quantifies the overall density of streamlines connecting GM regions throughout the brain and has been shown to predict domain-general cognitive performance[18] and lesion volume. CPM using MDN was marginally better than the edge density map in the cross-validation test within TMT-1 (ΔAIC = −5.9, $p = 0.0503$, $R = 1.01$; Fig. 4c). The advantage became statistically significant as the level of generalization increased in the analyses (TMT-1 → TMT-2: ΔAIC = −11.2, $p = 0.0036$, $R = 1.02$; TMT-1 → Stroop: ΔAIC = −17.8, $p = 0.00014$, $R = 1.04$; Fig. 4c). CPM

using MDN also outperformed prediction using total lesion volume in all three analyses (cross-validation within TMT-1: ΔAIC = −13.1, $p = 0.0014$, $R = 1.02$; TMT-1 → TMT-2: ΔAIC = −14.2, $p = 0.0008$, $R = 1.03$; TMT-1 → Stroop: ΔAIC = −34.6, $p = 3 \times 10^{-8}$, $R = 1.08$; Fig. 4c).

Lastly, we compared CPM using MDN to predictions based on lesion location without disconnection information (conducted using multivariate lesion-symptom mapping with lesion-symptom map). Two lesion-symptom map-based predictions were tested. First, we computed a lesion-symptom map-based on the TMT-1 sample and used it to predict behavioral scores in the TMT-2 and the Stroop samples separately. Both tests showed better performance for CPM using MDN (TMT-1 → TMT-2: ΔAIC = −7.8, $p = 0.020$, $R = 1.01$; TMT-1 → Stroop: ΔAIC = −25.2, $p = 3 \times 10^{-6}$, $R = 1.06$; Fig. 4g). Second, we performed the CPM on the collection of HCP cortical ROIs that overlapped the LESYMAP above (23% of all ROIs). A comparison of AC pattern maps between this set of ROIs and the ROIs included in the MDN is visualized in Supplementary Fig. 3. This CPM numerically outperformed the CPM on MDN in the analysis of TMT-1 → TMT-2 (ΔAIC = −4.5, $p > 0.095$, $R = 1.01$, Fig. 4g). However, it showed inferior predictive performance compared to the CPM on MDN when predicting Stroop performance using information from the TMT-1 sample (ΔAIC = −15.3, $p = 5 \times 10^{-4}$, $R = 1.03$; Fig. 4g). Together the results show that AC disconnection in the MDN predicted cross-task cognitive control performance better than a CPM that utilized results of lesion-symptom mapping.

### Comparison to other networks of cognitive control

A limitation of the current study is that the MDN is only one out of many possible iterations of cognitive control networks. As an exploratory analysis, we compared CPM performance on the MDN to

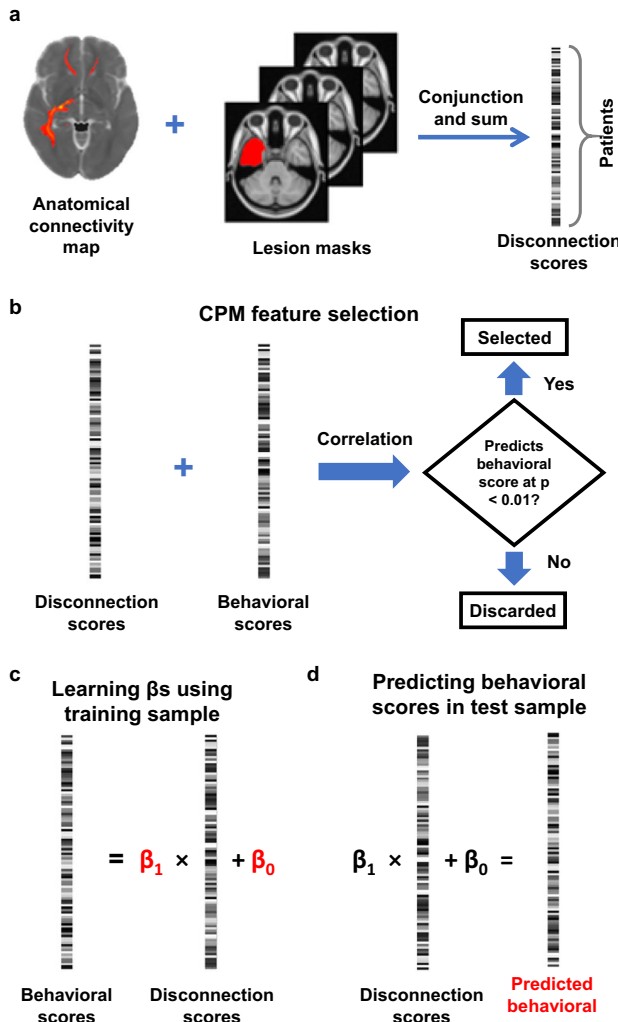

**Fig. 3 | Connectome-based predictive modeling (CPM) analysis overview.** This figure uses the AC disconnection score as an example. Analyses using functional connectivity (FC) disconnection scores follow the same procedure but use an FC map instead of anatomical connectivity (AC) map as input. **a** Computation of disconnection scores. For each cortical region of interest (ROI), an AC map is produced using deterministic tractography from a normative dataset. The extent of overlap of this AC map with each patient's lesion mask is quantified. Specifically, the voxel-wise sum of AC map values within the lesion mask is calculated and forms the patient's disconnection score for that specific cortical ROI. This process is repeated for each AC map derived from 360 cortical ROIs. **b** Feature selection of disconnection scores in CPM. A feature (i.e., disconnection scores for an ROI) is selected if it is positively correlated with behavioral scores with uncorrected $p < 0.01$ (tested using two-sided Pearson's correlation). **c** For a selected feature, disconnection scores are regressed against behavioral scores of the training sample to obtain regression coefficients (in red). The coefficients are the output of learning. (**d**) the coefficients are applied to disconnection scores from a test sample to make predictions of behavioral scores (in red), which were then compared to actual behavioral scores to assess prediction performance.

CPM performance on each of three networks that are involved in cognitive control and contribute to regions within the MDN, namely the frontoparietal network, the cingulo-opercular network and the dorsal attention network (Table 3). After FDR correction, the dorsal attention network showed better performance than the MDN in the TMT-1 → TMT-2 analysis ($\Delta$AIC = −7.2, $p$ = 0.026, $R$ = 1.01). However, MDN outperformed the dorsal attention network in the TMT-1 → Stroop analysis ($\Delta$AIC = −10.4, $p$ = 0.0055, $R$ = 1.02). No other networks were significantly better than MDN in predicting out-of-sample

performance after FDR correction. The overall similar prediction performance may be partly attributed to the overlap of the networks and the MDN. More research is needed to further delineate the neuroanatomy of the cognitive control network.

### Frontoparietal WM tracts predict cognitive control performance

To visualize the feature selection results of the CPM, Fig. 5a shows the cortical regions from which anatomical disconnection is predictive of impaired cognitive control (surface view of cortical ROIs color-coded according to predictive performance). In order to identify these disconnection patterns within the WM, we used FSL's permutation analysis of linear models (PALM, see the "Methods" section). Each cortical ROI's brain-wide AC map was used as the independent variable and the $T$-statistic of each cortical region (shown in Fig. 5a) was used as the dependent variable. In so doing, we can identify WM voxels that show stronger connectivity (i.e., having higher streamline values) with cortical ROIs whose AC disconnection scores are more strongly associated with cognitive control performance. In other words, disconnection in these WM voxels will be selectively linked to cognitive control performance. The results for TMT (TMT-1 and TMT-2 combined) and Stroop scores were shown in Fig. 5b and c, respectively. Both test scores relied on LH WM tracts linking frontal and parietal areas. When contrasting the two tests, Stroop appeared to depend more on frontal and posterior temporal WM tracts, whereas TMT showed a stronger reliance on both frontoparietal and occipito-temporal WM tracts (Fig. 5d). Importantly, both tests overlap in frontoparietal WM tracts (Fig. 5e), suggesting a shared WM correlate for different cognitive control tasks. The shared WM tracts (green in Fig. 5e) further overlap the AC map of the MDN (Fig. 5f). To test whether the volume of overlap is significantly above chance, we performed a random permutation test by randomly shifting the locations of the shared WM clusters (green in Fig. 5e) and computing its overlap volume with the AC map of the MDN across 1000 iterations. The overlap volumes thus from randomization formed a null distribution. The volume of overlap from Fig. 5f was significantly higher than the median of the null distribution ($p < 0.001$).

### Discussion

Cognitive control guides goal-directed behavior and is a hallmark of human intelligence. We aimed to identify the WM correlates underlying cognitive control by predicting cognitive control performance based on the location and connectivity patterns of acquired focal lesions. To comprehensively evaluate prediction performance, prediction outcomes were analyzed at three different levels of abstraction: within-sample cross-validation, cross-sample generalization, and cross-task generalization. Consistently across the three analyses, we observed four main results: (1) AC disconnection scores outperform FC disconnection scores and lesion-symptom maps in predicting cognitive control performance, (2) AC disconnection scores in the LH show better performance than RH, (3) AC connecting the MDN shows similar or better prediction performance compared to the whole-brain AC data, and (4) WM tracts connecting frontal and parietal areas are more strongly connected to cortical areas whose AC patterns better predict cognitive control performance.

When predicting cognitive control performance, we employed disconnection scores that inferred the connectivity strength of each lesion location with multiple cortical ROIs. This approach captures the communicative functions of the WM. These results add to the literature in at least four important ways.

First, our results are consistent with other recent work demonstrating better prediction performance using AC-based than FC-based disconnection scores (Fig. 4a) and lesion-symptom maps (Fig. 4g)[18,26–28]. Our findings extend this observation to the field of cognitive control. One possible explanation is that GM lesions are

**Table 2 | Model performance (in AIC) and confidence interval (in parentheses) from cross-validation within TMT-1, TMT-1 → TMT-2 generalization, and TMT-1 → Stroop generalization**

| | | Both hemispheres | Left hemisphere | Right hemisphere |
|---|---|---|---|---|
| *TMT-1: Cross-validation* | | | | |
| All ROIs | AC | −86.8 (−89.2, −82.8) | −87.4 (−89.7, −83.7) | −72.7 (−76.2, −66.6) |
| | FC | −68.2 (−73.6, −60.8) | −68.1 (−73.1, −61.4) | −74.5 (−77.5, −69.5) |
| | AC + FC | −86.2 (−88.7, −82.3) | −86.3 (−88.8, −82.4) | −75.3 (−78.4, −71.3) |
| Multiple demand network | AC | −83.1 (−85.6, −78.9) | −83.2 (−85.5, −79.0) | −68.5 (−74.6, −60.5) |
| | FC | −62.7 (−70.5, −55.4) | −62.2 (−69.8, −55.7) | −74.5 (−77.0, −69.6) |
| | AC + FC | −82.5 (−85.1, −78.5) | −82.3 (−84.9, −78.2) | −75.2 (−77.8, −70.8) |
| Baseline | Edge density | −77.3 (−80.3, −73.7) | N/A | N/A |
| | Lesion volume | −70.0 (−73.5, −65.7) | N/A | N/A |
| *TMT-1 → TMT-2* | | | | |
| All ROIs | AC | −160.4 (−234.0, −94.6) | −160.4 (−237.0, −106.4) | −148.4 (−216.1, −78.8) |
| | FC | −144.2 (−212.3, −76.8) | −143.3 (−215.6, −92.7) | −144.1 (−209.0, −66.3) |
| | AC + FC | −159.8 (−232.6, −92.4) | −160.4 (−237.1, −106.1) | −149.1 (−214.8, −72.0) |
| Multiple demand network | AC | −157.1 (−233.8, −91.5) | −156.4 (−231.9, −101.6) | −150.5 (−221.0, −79.5) |
| | FC | −146.4 (−211.6, −79.8) | −139.8 (−214.6, −89.1) | −146.4 (−208.6, −70.9) |
| | AC + FC | −158.7 (−233.0, −91.3) | −156.8 (−232.7, −101.8) | −150.0 (−213.8, −74.5) |
| Baseline | Edge density | −145.9 (−223.0, −85.4) | N/A | N/A |
| | LSM | −151.0 (−217.2, −101.0) | N/A | N/A |
| | LSM + CPM | −161.6 (−201.1, −154.1) | N/A | N/A |
| | Lesion volume | −142.9 (−214.1, −83.7) | N/A | N/A |
| *TMT-1 → Stroop* | | | | |
| All ROIs | AC | 123.6 (82.1, 157.5) | 121.8 (79.0, 150.0) | 162.8 (134.9, 185.4) |
| | FC | 152.3 (109.0, 181.9) | 147.3 (112.8, 181.5) | 154.5 (114.6, 183.8) |
| | AC + FC | 125.2 (83.1, 165.1) | 120.7 (77.3, 150.5) | 155.0 (113.6, 182.4) |
| Multiple demand network | AC | 111.4 (67.7, 149.2) | 109.8 (64.6, 138.9) | 157.0 (129.3, 183.4) |
| | FC | 160.0 (119.8, 188.6) | 131.3 (89.3, 170.6) | 165.0 (131.4, 191.3) |
| | AC + FC | 118.8 (74.0, 159.4) | 105.8 (59.0, 135.7) | 165.2 (129.8, 190.8) |
| Baseline | Edge density | 129.2 (97.6, 157.0) | N/A | N/A |
| | LSM | 135.4 (73.0, 219.5) | N/A | N/A |
| | LSM + CPM | 126.6 (81.3, 162.1) | N/A | N/A |
| | Lesion volume | 145.9 (116.1, 171.7) | N/A | N/A |

N/A: Not applicable. LSM + CPM: CPM performed on ROIs defined by lesion-symptom map results on TMT-1. Bold font indicates the analysis conducted.

better compensated by plasticity mechanisms relative to WM lesions[28,29], and thus the location is less predictive of behavioral outcomes long after the lesion.

Second, our results suggest LH disconnection scores predict cognitive control performance better than the RH, a finding consistent laterality differences observed in brain activity while performing cognitive control tasks[30,31]. Note that due to the limited tests of cognitive control in our data, inhibitory control (i.e., inhibiting a potent response or canceling initiated action), which involves the right inferior frontal cortex[32], was not tested in this study. Thus, future research needs to include more neuropsychological tests of cognitive control to investigate both generic predictors capturing domain-general cognitive control and predictors specialized for individual cognitive control types (i.e., domain-specific cognitive control). In our experimental design, the TMT and the Stroop task share cognitive control processes that inhibit the processing of distractors (e.g., non-target letters and digits in the TMT and the word meaning in the color–word condition of the Stroop task). Inhibition is believed to be a key component of domain-general cognitive control[33]. These shared processes may be reflected in the overlap of the lesion-symptom maps (Fig. 2b) and the overlap of cortical ROIs whose AC disconnection scores significantly scaled with TMT and Stroop measures (Fig. 5a). On the other hand, the TMT and the Stroop tasks also rely on different, domain-specific cognitive control processes. For example, the TMT part B involves

cognitive flexibility that switches search targets between letters and digits, in contrast to the sustained, stable cognitive control on the color–word interference portion of the Stroop task. Cognitive stability and cognitive flexibility are considered different aspects of cognitive control[34]. Additionally, the two tasks have distinct targets of cognitive control: the TMT engages cognitive control to modulate visual search, whereas the Stroop task requires cognitive control to modulate stimulus-response associations[35]. Indeed, the co-existence of brain regions supporting domain-general and domain-specific cognitive control is a common finding in the literature[20,21] and is typically documented as partially overlapping networks, each of which represents cognitive control in a specific domain. We speculated the partially overlapping sets of brain areas for the TMT and the Stroop measures in Figs. 2b and 5a may capture the co-existence of domain-general and domain-specific cognitive control.

Our third main finding is that prediction performance using disconnection scores from the MDN, which is ~17% of the 360 cortical ROIs, is similar to or better than the performance using all ROIs' disconnection scores (Fig. 4c). This is consistent with theories and empirical evidence that cognitive control is mainly supported by a network of selective cortical and subcortical regions[8,9]. Although there is diversity in how the network is defined and what brain areas are included in this network, a general consensus is that prefrontal and (inferior) parietal regions are involved in the network to implement

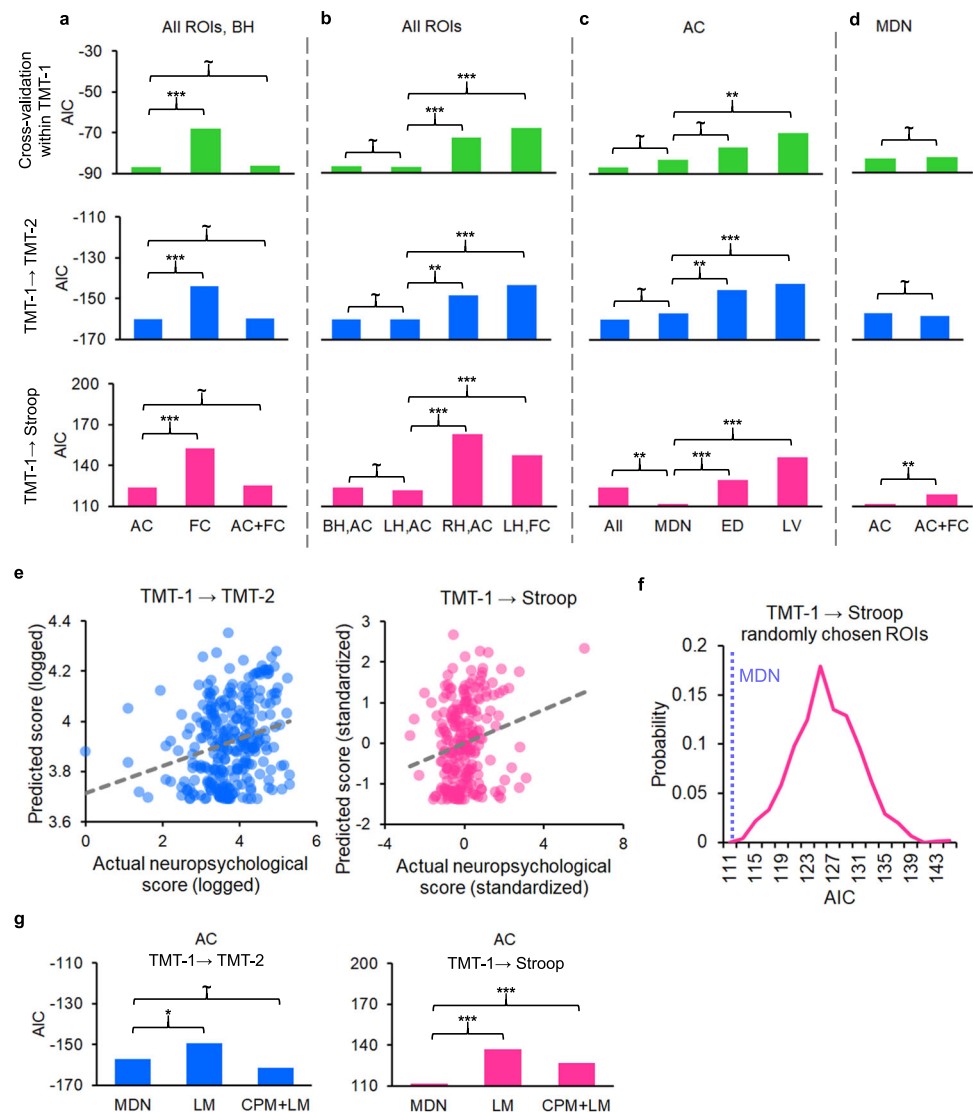

**Fig. 4 | Performance of predicting cognitive control performance using lesion data across participants. a–d, g** Prediction performance, quantified using Akaike information criterion (AIC), plotted as a function of the collection of connection scores used in prediction. *p*-values are from one-sided tests of difference in AIC (see the "Methods" section). **a** Comparison between anatomical connectivity (AC) and functional connectivity (FC) prediction performance using all regions of interest (ROIs). AC outperformed FC in all three analyses (cross-validation within trail-making test-1 [TMT-1]: $p = 9 \times 10^{-5}$; TMT-1→TMT-2: $p = 0.0003$; TMT-1→Stroop: $p = 6 \times 10^{-7}$). **b** Lateralization of prediction performance using all ROIs. AC disconnection scores in the left hemisphere (LH) were better predictors than those in the right hemisphere (RH, cross-validation within TMT-1: $p = 0.0007$; TMT-1→TMT-2: $p = 0.0024$; TMT-1→Stroop: $p = 1 \times 10^{-9}$). LH AC disconnection scores consistently outperformed LH FC disconnection scores in predicting cognitive control performance (cross-validation within TMT-1: $p = 7 \times 10^{-5}$; TMT-1→TMT-2: $p = 0.0002$; TMT-1→Stroop: $p = 3 \times 10^{-6}$). **c** Performance using multiple demand network (MDN) AC disconnection scores compared to that of AC disconnection scores from all ROIs, edge density map, and lesion volume. When tested between the TMT and the Stroop task, MDN yielded better prediction performance than all ROIs ($p = 0.0022$). Connectome-based predictive modeling (CPM) using MDN was

at least marginally better than the edge density map (cross-validation test within TMT-1: $p = 0.0503$; TMT-1→TMT-2: $p = 0.0036$; TMT-1→Stroop: $p = 0.00014$). CPM using MDN also outperformed prediction using total lesion volume in all three analyses (cross-validation within TMT-1: $p = 0.0014$; TMT-1→TMT-2: $p = 0.0008$; TMT-1→Stroop: $p = 3 \times 10^{-8}$). **d** Performance using MDN AC disconnection scores compared to performance using both MDN AC and FC disconnection scores. Performance is better using only AC disconnection scores than both AC and FC disconnection scores when tested on Stroop data ($p = 0.0024$). **e** Scatter plots showing the relation between predicted and actual neuropsychological scores in cross-sample and cross-task generalization tests using MDN AC disconnection scores. Trend lines were plotted as dashed gray lines. **f** Probability density function of AICs from 1000 randomly selected ROI sets that have the same number of ROIs as the MDN. **g** Prediction performance using CPM on MDN, compared to that of a lesion-symptom map from TMT-1 (lesion-symptom mapping [LM]) and CPM with cortical ROIs overlapping the lesion-symptom map (CPM + LM). When tested on TMT-2 data, CPM on MDN outperformed LM ($p = 0.020$). When tested on Stroop data, CPM on MDN outperformed both LM ($p = 3 \times 10^{-6}$) and CPM + LM ($p = 5 \times 10^{-4}$). -: $p > 0.05$; *: $p < 0.05$; **: $p < 0.01$; ***: $p < 0.001$. BH both hemispheres, ED edge density, LV lesion volume. Source data are provided as a Source Data file.

cognitive control as prefrontal top-down modulation of posterior processing[2]. Supporting this view, our last main result shows that disconnection in WM tracts connecting frontal and parietal areas is predictive of impaired cognitive control performance (Fig. 5d). Similarly, fMRI connectivity analysis has shown that neural activity in posterior visual areas is modulated by prefrontal activity as a function

of the strength of cognitive control[36], which also suggests connectivity between frontal and posterior cortical areas is crucial in supporting cognitive control. Importantly, lesions in the frontoparietal WM tracts are associated with impairments in both neuropsychological tests of cognitive control (Fig. 5e). This result suggests shared WM correlates between the two tasks involving cognitive control and may reflect that

**Table 3 | AC-based CPM performance comparison to the MDN**

| Network | Analysis | AIC (95% CI) | ΔAIC to MDN | p-value (uncorrected) | R |
|---|---|---|---|---|---|
| Frontoparietal network | Within TMT-1 | −81.6 (−84.0, −76.9) | 1.6 | 0.312 | 1.00 |
| | TMT-1 → TMT-2 | −153.4 (−219.1, −84.0) | 3.6 | 0.140 | 1.00 |
| | TMT-1 → Stroop | 117.7 (70.4, 154.8) | 6.4 | 0.040 | 1.01 |
| Cingulo-opercular network | Within TMT-1 | −78.5 (−81.5, −73.4) | 4.6 | 0.090 | 1.01 |
| | TMT-1 → TMT-2 | −155.9 (−220.2, −86.7) | 1.2 | 0.358 | 1.00 |
| | TMT-1 → Stroop | 105.2 (56.2, 146.1) | −6.2 | 0.044 | 1.01 |
| Dorsal attention network | Within TMT-1 | −86.0 (−88.6, −81.5) | −2.9 | 0.188 | 1.00 |
| | TMT-1 → TMT-2 | −164.3 (−231.1, −93.2) | −7.2 | **0.026** | 1.01 |
| | TMT-1 → Stroop | 121.7 (77.4, 156.6) | 10.4 | **0.0055** | 1.02 |

p-Values are from AIC comparison (one-sided tests, see the "Methods" section). Bold font indicates p-values survived FDR correction. A negative change in AIC reflects better performance and higher positive values reflect worse performance.

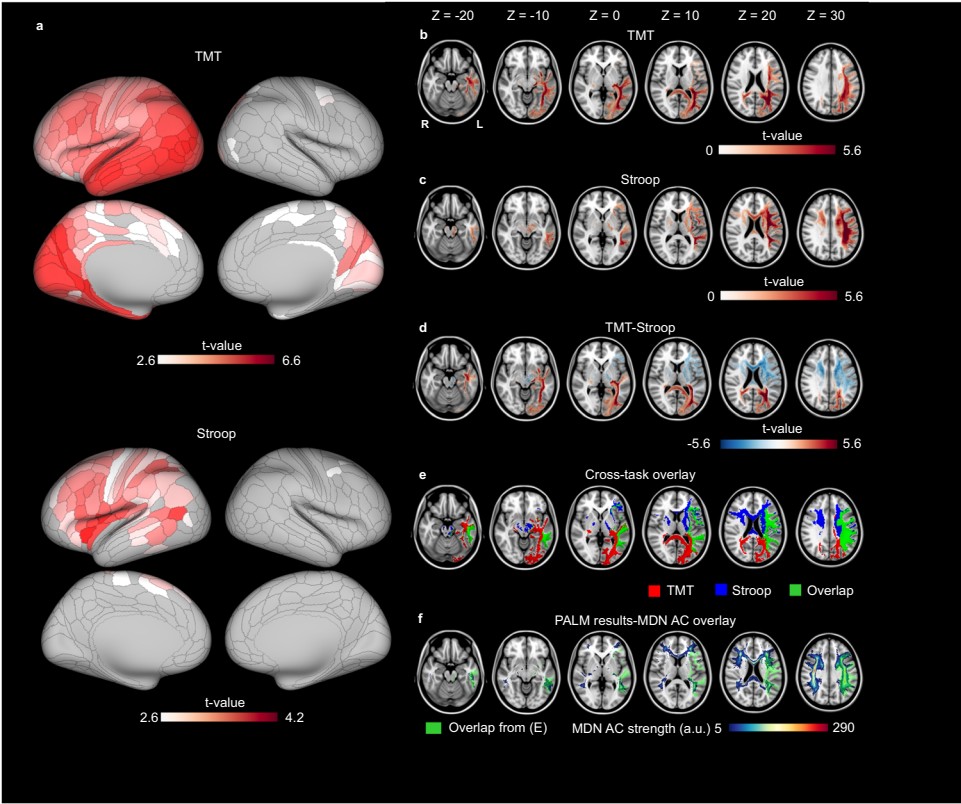

**Fig. 5 | Permutation analysis of linear models (PALM) analysis results. a** Cortical regions of interest (ROIs) whose anatomical connectivity (AC) disconnection scores significantly scaled with cognitive control impairment (false-discovery rate [FDR] corrected). **b** Results of trail-making test (TMT) scores. For each voxel, the color indicates the *t*-value of the PALM analysis. Only voxels that survived multiple comparison corrections (two-sided *p* < 0.05 corrected) are colored. **c** Results of Stroop scores. Color coding is identical to (**b**). **d** *t*-value difference between (**b**) and (**c**). **e** Overlap of (**b**) and (**c**). **f** White matter (WM) tracts shared by TMT and Stroop (in green) and its overlap with multiple demand network (MDN) AC map.

the frontoparietal tracts are part of the domain-general cognitive control mechanisms[19] and/or that the WM tracts connect brain regions that adaptively encode information for different tasks[10]. We further found that the WM tracts shared by the two tasks significantly overlapped the AC map of the MDN (Fig. 5f). Consistent with the MDN's proposed function of supporting performance in highly demanding cognitive control tasks, CPM limited to the MDN cortical regions identified white matter regions with robust cross-task prediction of cognitive control performance (Fig. 4c). On the other hand, when lesion disconnection scores from all ROIs were used, WM tracts that are only sensitive to one task (e.g., supporting domain-specific cognitive control, Fig. 3c and d) may be selected in the predictive model, leading to overfitting and worse prediction performance than MDN in

the cross-task analysis (Fig. 5c). This may also explain the finding that AC-based CPM defined on lesion-symptom map results performed worse than MDN AC-based CPM (Fig. 4g) when predicting Stroop task performance using performance in TMT-1, as lesion-symptom mapping method may have relied on neural correlates specific for TMT to predict performance in the Stroop task. To fully investigate the neural substrates of domain-general and domain-specific cognitive control, more tasks capturing cognitive control and more data to further increase lesion coverage are needed.

CPM is a powerful statistical approach that has been applied to neuroimaging data to account for individual differences[37] and predicting behavioral scores of sustained attention[24]. The power of the method partly comes from the boosting technique[38], or combining

individual predictors to increase prediction performance. We applied the CPM approach to a lesion database. Specifically, we calculated disconnection scores for each lesion to 360 cortical regions. Cortical region disconnection scores that were significantly associated with cognitive control performance in the training data were used to generate regression coefficients, which were then used to predict cognitive control performance in independent samples (Fig. 3). The selected cortical regions leverage the boosting technique by incorporating multiple individual predictive variables and, thus, outperforming methods that would rely on only one or a small number of regions. Further, we have proposed an approach for back-projecting those results to the anatomical regions containing white matter tracts where disconnection is most predictive of impaired performance.

In summary, we found that AC maps linking the MDN, in particular left frontoparietal WM tracts, can robustly predict cognitive control performance following acquired brain lesions. The findings provide evidence of the WM correlates underlying cognitive control and have the potential to forecast a deficit in cognitive control performance following brain lesions.

## Methods

### Participants

Six hundred and forty-three patients with focal brain lesions participated in this study, including 564 from the Iowa Cognitive Neuroscience Patient Registry and 79 from the Benton Neuropsychology Clinic. Prior to any data collection, this study was approved by the Institutional Review Board of the University of Iowa. Participants recruited through the Patient Registry signed informed consent prior to performing cognitive testing while de-identified clinical data from participants from the Benton Neuropsychology Clinic were acquired by chart review of retrospective electronic medical records following approval by the Institutional Review Board. Six hundred and twenty-two patients completed the TMT and 229 patients completed the Stroop task (208 patients completed both tasks). Participant sampling and testing procedures are detailed in Supplementary Note 5. Participants were divided into three groups based on the experimental design. Specifically, group 1 contains Iowa Cognitive Neuroscience Patient Registry patients that only completed the TMT. The remaining patients with TMT scores were included in group 2. The Stroop sample contains all patients that completed the Stroop task and were not included in group 1. The partitions ensured data independence in the generalization tests (see below) while balancing the number of subjects between samples. The demographic information for each sample is listed in Table 1.

### Behavioral assays

We utilized two different tasks to assess cognitive control performance: TMT and Stroop task. TMT consisted of parts A and B. In part A (TMT A), participants were required to link 25 circles, each of which contains a number from 1 to 25, by drawing lines in ascending order of number values as fast as possible. In part B (TMT B), participants encountered circles containing either a number or a letter and were required to link the circles by drawing lines in ascending order, and at the same time, alternating between number and letter circles (1-A-2-B-3-C, etc.), as fast as possible. While TMT A and TMT B both measure motor skills and basic visual attention function, TMT B is relatively more difficult and requires higher cognitive control demand, including working memory and task switching. Behavioral scores were measured by time spent (in seconds) in each of the two parts. We subtracted TMT A scores from TMT B scores (TMT B-A) for each participant to measure cognitive control-specific impairment while minimizing performance effects related to functions other than executive functions (e.g. hemiparesis or visual field deficits), as performed previously[39]. To reduce the impact of outliers, TMT B-A scores were log-transformed.

The Stroop test consisted of three different conditions: color (C), word (W), and color-word (CW). In the C condition, patients were presented squares painted in either red, blue, or green color and were asked to name the color of the squares. In the W condition, patients were asked to read the words that are written in black ink, and words could be either 'red', 'blue', or 'green'. Lastly, in the CW condition, patients saw the words 'red', ''blue', and 'green', but printed in incongruent colors (e.g., the word 'red' in green ink color), and are required to state the color of the ink for the word. The CW condition is more cognitively demanding than other conditions as cognitive control is required to suppress the habitual response of reading the word and to select the response of naming the ink color. In each condition, participants were given 45 s, and the number of correct responses was recorded. Similar to the TMT, we focused on response times (i.e., seconds per correct response) and calculated the interference scores as CW−(C + W)/2. Interference scores were then log-transformed.

### Lesion segmentation

Each participant included in the analysis had a focal brain lesion with visible boundaries evident from structural imaging sequences on MRI. CT scans were used in rare cases when MRI was contraindicated ($n = 64$). Lesions were manually segmented in three dimensions by a rater blind to behavioral test scores, and the anatomical accuracy of each tracing was reviewed by a neurologist (A.D.B.) in both native space and upon transformation to MNI152 1-mm template brain using a combination of linear and nonlinear registration techniques, as performed previously[40].

### Multivariate lesion-symptom mapping

Lesion-symptom mapping (LESYMAP) analyses were performed on the TMT and Stroop results using sparse canonical correlation analysis (SCCAN) as implemented in LESYMAP[22], a package available in R (https://github.com/dorianps/LESYMAP). The SCCAN method involves an optimization procedure that finds voxel weights that maximize the multivariate correlation between voxel values and behavioral scores. The predictive value and sparseness of the model are derived empirically using a 4-fold, within-sample correlation between model-predicted and actual behavioral scores. LESYMAP deems a map "valid" if it is associated with a statistically significant predictive correlation. Briefly, SCCAN builds a model using 75% of the sample, applies it to the remaining 25% of the sample in order to predict the scores from lesion location, and correlates these predictions with actual scores. Thus, this approach tests the statistical significance of the entire map at once and avoids the pitfalls associated with voxel-wise (i.e., mass univariate) methods, such as inflated rates of false-positive errors. This previously validated method has been demonstrated as more accurate than mass univariate methods and is better able to identify when multiple brain regions are associated with a behavioral variable.

### CPM ROI definitions

We divided the whole cortical space into 360 ROIs (180 per hemisphere) based on the HCP (HCP-MMP1.0) atlas[23]. All 360 ROIs were used for whole-brain CPM analysis. When the CPM analysis was restricted to the multiple demand network, the regions identified in Assem et al.[8] were used.

### Connectome-based predictive modeling

In this study, CPM[24,25] was used to evaluate how lesion location predicts behavioral outcomes (i.e., TMT B-A, Stroop interference). In brief, CPM consists of a feature selection and a prediction stage. In the feature selection stage (Fig. 3b), the overlap of a lesion with an underlying anatomical map of interest is computed as the lesion disconnection score and was then regressed against the behavioral score (e.g., TMT B-A). The ROI is selected for inclusion in the model if its uncorrected

*p*-value of the regression (relating lesion disconnection scores and behavioral performance) is lower than 0.01[24]. We used ROI inputs into CPM that spanned functional connectivity and structural connectivity measures (described below). For AC scores, we further restricted the feature selection to lesion load values with positive correlation (i.e., larger lesion load is associated with worse performance). In the prediction stage, each selected ROI's disconnection scores were regressed against the behavioral scores separately across participants in the training sample. The regression coefficients were used to predict behavioral scores in the test sample (Fig. 3d). The final prediction is the average of each selected disconnection score's prediction.

### AC disconnection scores

We used the DSI studio/LEAD-DBS Connectome pipeline (https://www.lead-dbs.org/about/lead-connectome/[41]) to explore the WM tracts associated with each of the HCP atlas ROIs. As performed previously[42], this software uses normative diffusion-weighted MRI from 32 subjects from the HCP dataset to perform deterministic fiber tractography and estimate the course of WM streamlines associated with a seed region-of-interest[43] (https://ida.loni.usc.edu/). This data was selected as the state-of-the-art sequences and specialized hardware make it one of the best openly available, in-vivo diffusion MRI datasets available[44]. Diffusion-weighting with $b = 1000$ and 3000 s/mm$^2$ was applied along 64 directions. Furthermore, additional shells at *b*-values of 5000 and 10,000 s/mm$^2$ were applied in 128 directions. In-plane resolution was $1.5 \times 1.5$ mm and 96 slices with a 1.5 mm thickness were acquired. The voxel-wise whole brain maps containing the streamlines from each seed ROI were used to generate the AC disconnection scores. This was defined as the voxel-wise sum of each ROI's WM connectivity map masked by each patient's lesion map. As the AC maps were generated using an independent dataset, they are not confounded by the lesions in the subjects.

### FC disconnection scores

For each ROI, its FC network was constructed using rs-fcMRI. The primary rs-fcMRI dataset included 98 healthy right-handed subjects (48 male subjects, age $22 \pm 3.2$ years), that were resting quietly at the time of data collection. These data are part of a larger, publicly available data set used previously[45–47]. Rs-fcMRI data were processed in accordance with previously described methods[48–51]. Participants completed two 6.2 min rs-fcMRI scans during which they were asked to rest in the scanner (3 T, Siemens) with their eyes open (TR = 3000 ms, TE = 30 ms, FA = 85, 3 mm voxel size [27 mm$^3$], FOV = 216, 47 axial slices with interleaved acquisition and no gap). Functional data were spatially smoothed using a Gaussian kernel of 4 mm full-width at half-maximum. The data were temporally filtered (0.009 Hz $< f < 0.08$ Hz) and several nuisance variables were removed by regression, including the following: (a) six movement parameters computed by rigid body translation and rotation during preprocessing, (b) mean whole-brain signal[52,53], (c) mean brain signal within the lateral ventricles, and (d) the mean signal within a deep WM ROI. The inclusion of the first temporal derivatives of these regressors within the linear model accounted for the time-shifted versions of spurious variance. For each voxel, linear regression with the above model was applied to its fMRI signal time course. The residual of the regression was used to measure FC, which was defined as the correlation of residual fMRI signal time course between the mean of the ROI and each voxel. Correlation coefficients were converted to normally distributed *Z*-scores using the Fisher transformation and group-averaged results were reported as voxel-wise *Z*-scores. As a result, a FC connectivity map was built for each ROI, containing a voxel-wise *Z*-score between the voxel and the average fMRI signal time course of the ROI. Finally, for each participant and each ROI, the FC disconnection score was computed as the sum of the *Z*-scores in the ROI's FC connectivity map masked by the participant's lesion map.

### Statistical analysis of prediction performance comparison

For each model (e.g., AC from all ROIs), the AIC was computed using the predicted and actual behavioral scores. Assuming two models $M_1$ and $M_2$ with AICs of $AIC_1$ and $AIC_2$, respectively, the likelihood of $M_1$ outperforming $M_2$ is $L = e^{(AIC_1 - AIC_2)/2}$. Subsequently, the *p*-value corresponding to the null hypothesis that $M_1$ does not outperform $M_2$ is $\frac{L}{L+1}$. FDR correction[54] was applied across the three analyses (i.e., within TMT-1, TMT-1 → TMT-2, and TMT-1 → Stroop).

To estimate the confidence interval (CI), we used bootstrapping to randomly sample the participants with replacement. Each random sample had the same sample size as the original sample. For each statistic, one hundred random samples were generated to estimate the 95% CI.

To estimate the effect size, for each model we first computed its average prediction error $\hat{\sigma}_e = \sqrt{\frac{\sum(y - \hat{y})^2}{n}}$, where n is the sample size, and $y$ and $\hat{y}$ are actual and model prediction of behavioral scores, respectively. For the effect size of a model comparison, we reported the ratio of $\hat{\sigma}_e$, denoted as *R*. We always use the better model (i.e., model with lower average prediction error) as the denominator, such that the effect size is no less than 1. The effect size quantifies on average how much larger the prediction error of the worse model is than the better model. For example, a ratio $R = 1.03$ indicates that on average the worse model produces 3% more prediction error than the better model for each patient.

### Identifying structural disconnection anatomy in CPM analysis

In order to identify the regional WM findings associated with the CPM we used an FSL-based voxel-wise permutation analysis of linear models (PALM, https://fsl.fmrib.ox.ac.uk/fsl/fslwiki/PALM). The 360 unthresholded individual tractography maps derived from each cortical ROI in the CPM were used to provide voxel-level predictors in a general linear model. The dependent variable of the model was the *T* score from the feature selection stage of the CPM. The general linear model was used to identify white matter regions significantly associated with ROIs with better performance (i.e., higher *T* scores) in predicting cognitive control performance. Statistical significance was evaluated with threshold-free cluster enhancement, 2-tailed significance, and 2000 permutations. Regional findings were compared to the HCP-842 and JHU white matter tractography atlases to relate our findings to common white matter tracts[55–59].

### Reporting summary

Further information on research design is available in the Nature Portfolio Reporting Summary linked to this article.

## Data availability

Lesion masks and neuropsychological test data are available under restricted access, subject to the policies and procedures of the Iowa Cognitive Neuroscience Patient Registry and the Benton Neuropsychology Clinic. Data access may be requested by emailing A.D.B. and will be replied within a month. Source data are provided with this paper.

## Code availability

Analysis code and sample data is available at https://github.com/JiefengJiang/CPMCognitiveControl[60].

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

## Acknowledgements

This project was supported by the National Institute of Mental Health (R01MH131559 to J.J.) and the National Institute of Neurological Disorders and Stroke (R01NS114405 to A.D.B.).

## Author contributions

Conceptualization: J.J., W.L., and A.D.B.; Methodology: J.J., J.B., W.L., and A.D.B.; Software: J.J. and J.B.; Formal analysis: J.J., J.B., W.L., and A.D.B.; Data curation: J.B., D.T., and A.D.B; Writing original draft: J.J.; Writing, review, and editing: J.B., W.L., D.T., and A.D.B.; Funding: J.J. and A.D.B.; Supervision: J.J., A.D.B., and D.T.

## Competing interests
The authors declare no competing interests.
