## [Peer Review File · Nature Communications]

White matter disconnection of left multiple demand network is associated with post-lesion deficits in cognitive controlReviewer #1 (Remarks to the Author):

This study of a large cohort of focally-lesioned patients sets out to test the hypothesis that disconnection of MDN predicts TMT and Stroop task performance. Rejecting the null hypothesis is interpreted as supporting the conclusion that the MDN is necessary for cognitive control. The results are interesting, but several aspects of the design and execution are problematic.

1. The study is presented as an investigation into cognitive control, but is really an investigation into the lesion-symptom relations of TMT and Stroop, and prediction of TMT and Stroop scores from TMT data alone. These tasks require more than just cognitive control, and though adjusted scores are used there is no contextual neuropsychological data to reassure us there are no confounding effects from other domains. Some patients received TMT, others also Stroop, with no indication of how allocation was determined or is potentially biased. Very little behavioural analysis is presented, even of the tasks themselves. This all makes it difficult to tell what the authors are really mapping.

2. A large number of lesions are analysed but nothing is said about the underlying clinical diagnosis. Were patients with trauma included? The lesion distribution shows a strong frontal preponderance. This reinforces the concern that the sample may be biased.

3. The hypothesis that disconnection of MDN predicts Stroop & TMT performance is primarily tested by comparing the performance of disconnection models including the MDN vs a random selection of a comparable number of ROIs. But given that predictive performance is poor (see Figure 4) a comparison of models is only weak evidence for spatial specificity. If lesion volume or lesions analysed without disconnection produce comparable performance, the argument for specificity becomes hard to sustain. These null models are not provided.

4. It is not clear why the authors do not compare the lesion-symptom map against the MDN map: the degree of overlap is a direct indicator of the anatomical dependence of the function they are studying. Does a model of the Lesymapped regions, either of lesions or disconnections, predict scores better than MDN out of sample? If it does, then the network that best predicts TMT and Stroop performance is not the MDN but another set of regions, even if MDN regions might be part of it.

5. The minimal degree of overlap between TMT and Stroop casts doubt on the claim that the two tests are measuring the same cognitive function, within the present experimental setup.

Reviewer #2 (Remarks to the Author):

This is a very important study that should make a valuable contribution to the literature. The Iowa patient cohort is one the best characterized patient cohorts in the world and provides an opportunity to answer questions about brain-behavior relationships that cannot be answered by other available patient cohorts. This study is well-designed, the methods are sound and innovative, the results are clearly presented and the conclusions drawn are supported by the results. However, there are a number of issues that I believe should be addressed that could potentially strengthen the manuscript. These are listed below in no particular order of importance.

(1) The etiology of the lesions should be described. The mechanisms of recovery/reorganization of function for tumor resections, traumatic brain injury, and stroke, likely differ. Although it is not clear that this issue affected the results of the study, the authors should acknowledge the heterogeneity of their patients.

(2) The length of time elapsed from the onset of injury to testing should be added to Table 1. Although it is not clear that this issue affected the results of the study, the authors should acknowledge this potential confound.

(3) There is a missed opportunity having not investigated other well-documented intrinsic frontal networks derived from resting fMRI data such as the frontoparietal network (FPN), dorsal attention network, ventral attention network, or cingulo-opercular network, all of which have frontal and parietal nodes. As the authors know, the MDN largely overlaps with the FPN network, but recently it has been suggested may also comprise the cingulo-opercular network. At the very least, the authors should make a stronger case in the introduction for only investigating the MDN, as well as note that other frontal networks should be investigated in the future. And in the introduction (line 75), the authors should not refer to the MDN, as “the cognitive control network” since it is one of many involved in cognitive control.

(4) The authors make this statement in the results section: “Importantly, the left dorsolateral prefrontal cortex and underlying WM is a region that, when lesioned, is associated with impairments in both TMT and Stroop performance”. Given how precise the anatomical data is in this study, the authors should be precise in their claims. What studies are the authors referring to that have demonstrated that a lesion in a portion of PFC (which looks like the posterior inferior frontal gyrus) causes deficits in the TMT and Stroop. All too often, studies make reference to the lateral PFC in a non-specific way, and this type of statement potentially permeates that type of impreciseness of discussion of lesion-behavior relationships.

(5) In general, I prefer that there is little speculation in I discussion sections. However, since this paper likely will open the eyes of many cognitive neuroscientists regarding the importance of white matter damage, the authors have an opportunity to inform the reader regarding why AC is more predictive of neuropsychological deficits than FC, and why white matter damage can be more predictive than cortical damage.

Reviewer #3 (Remarks to the Author):

I would like to thank the editor for the opportunity of reviewing this paper on the anatomical and functional disconnection associated with cognitive control deficits. The authors investigated the cognitive control abilities of a large dataset of stroke patients via two classic cognitive control tasks, the TMT and the Stroop task. To specifically explore the role of disconnection in the patients' performance, the patients' anatomical and functional disconnection profiles have been extracted and exploited into prediction models that predicted the performance of an independent sample. Interestingly, the authors implemented a cross-task validation together with a within-group and across-group validation. The results show high performance-prediction accuracy for anatomical connectivity compared to functional connectivity, in particular, left hemisphere connectivity. Moreover, the authors further compare the prediction performance of connectivity of the MDN with whole-brain connectivity demonstrating the role of white matter structures associated with cognitive control. The study is of potential interest to the readers of Nature Communications. However, a few methodological concerns need to be addressed.

- The sample size of the study is impressive. Nevertheless, crucial information on the patients is missing. Could the authors provide more detail on the neuropsychological and neurological status of the patients? For instance, the presence of co-occurring symptoms such as aphasia, neglect, fatigue, hemiplegia and slowness of reaction time could affect the performance in the tasks and hinder the association of the lesion profile to the deficit in cognitive control. Other information, such as the intervals between lesions and assessment/imaging and the lesion size, are missing and would be of

potential use in the analysis.

- What is the rationale for the overlap of the lesions onto the white matter structures associated with the MDM? What information does this analysis provide? Particularly when considering the more informative analyses conducted and described in the following paragraphs.

- I am puzzled by how the AC maps are calculated, as they might carry a bias intrinsic to the patient population. Specifically, the score refers to the number of patients who have a disconnection in that specific voxel, disregarding the lesion size and location. This might affect the following analyses. In fact, the successful prediction of the performance of another dataset could be explained by the lesion size and the most likely localisation of the stroke rather than only the goodness of the models. Why did the authors not consider using a more objective score (e.g. the number of streamlines of the healthy participants passing by the specific voxel)? This would allineate the anatomical score with the FC maps score derived from normative data.

- Other variables like lesion size, interval and co-occurring symptoms were not considered in the models. Could the authors comment on their choice and how they prevented these confounding factors from driving their predictions?

We thank the reviewers for their positive feedback and constructive comments. We have substantially revised the manuscript based on the reviewers' feedback and believe that the manuscript has improved greatly from its previous version. Please find below our point-by-point responses to the reviewers' comments.

Reviewer #1

0. This study of a large cohort of focally-lesioned patients sets out to test the hypothesis that disconnection of MDN predicts TMT and Stroop task performance. Rejecting the null hypothesis is interpreted as supporting the conclusion that the MDN is necessary for cognitive control. The results are interesting, but several aspects of the design and execution are problematic.

Response: We thank the reviewer for the helpful comments. In our responses below, we detail how additional analyses were performed to address the comments.

1a. The study is presented as an investigation into cognitive control, but is really an investigation into the lesion-symptom relations of TMT and Stroop, and prediction of TMT and Stroop scores from TMT data alone. These tasks require more than just cognitive control, and though adjusted scores are used there is no contextual neuropsychological data to reassure us there are no confounding effects from other domains.

Response: The reviewer raises several important points here that we address in the revised version. First, we added additional analyses of the behavioral data. We agree with the reviewer that it is important to address the construct validity of our measures and specifically their relationship to cognitive control. We added additional discussion and references supporting the use of the TMT and the Stroop measures as assessments of different aspects of cognitive control. In addition, we performed additional analysis on our data to test the discriminant validity between our measures of cognitive control and measures of other cognitive domains. Finally, we include this reliance on two tasks as estimates of cognitive control as a limitation, and a future direction of this research would be to include a more comprehensive assessment of cognitive control. The related text in the revised manuscript is quoted below:

“We define cognitive control as the top-down modulation on neural processing to achieve goal-directed behaviors, consistent with its use in the literature^{1,2}. Previous neuropsychological studies have shown that the TMT B-A scores^{3,4} and the Stroop interference scores⁵ capture cognitive control relatively cleanly, as they are contrast scores that attempt to control for inter-individual behavioral differences that rely on processes other than cognitive control that could also influence performance (e.g. motor control, understanding instructions, visual processing, processing speed). To assess the construct validity of these measures in this study, we took a discriminant validity approach. That is, we tested whether variability in TMT B-A and Stroop interference scores can be explained by classic measures of other cognitive processes. Specifically, we chose three measures, namely the vocabulary score from the 4th edition of the Wechsler Adult Intelligence Scale (WAIS-IV), the symbol search score from the WAIS-IV, and the Judgment of Line Orientation score⁶, as measures of verbal naming, processing and visuospatial abilities, respectively. The measures were chosen based on their availability in the database and their relevance to the TMT and Stroop tasks (i.e., the Stroop task involves words of color; the TMT requires subjects to search for letters and/or digits, similar to the symbol search task of the WAIS-IV; and both tasks require basic visual processing).

Note that the three selected measures may also be confounded by cognitive control when they are used to analyze individual differences. For example, higher cognitive control can enhance the processing of task-relevant information and improve the suppression of distractors (e.g., distractors from the task, other internal thoughts and/or the testing environment) and can thus benefit subjects in all tests. Thus, to test whether our measures are confounded by each of the three selected measures while controlling for potential shared variance related to cognitive control, for each selected measure (e.g., vocabulary score of the WAIS-IV) and each of the two cognitive control measures (e.g., TMT B-A), we performed a linear regression analysis using the former to predict the latter, while using the other cognitive control measure (e.g., Stroop interference score) as a nuisance predictor to control for shared variance due to cognitive control. None of the three selected measures displayed statistically significant effect in explaining variances in the two measures of cognitive control ($n = 72-204$, depending on the availability of test scores; all p s > 0.08 after FDR correction). The findings from the discriminant validity analysis suggest that the TMT B-A and the Stroop interference score cannot be explained by

impaired performance in these other related processes and are consistent with the results capturing performance related to cognitive control.” (Supplementary Note 1)

and

“Note that due to the limited tests of cognitive control in our data, inhibitory control (i.e., inhibiting a potent response or cancelling initiated action), which involves the right inferior frontal cortex³⁰, was not tested in this study. Future research needs to include more neuropsychological tests of cognitive control to investigate both generic predictors capturing domain-general cognitive control and predictors specialized for individual cognitive control types (i.e., domain-specific cognitive control).” (Page 15)

***Ib.** Some patients received TMT, others also Stroop, with no indication of how allocation was determined or is potentially biased.*

Response: To avoid potential confusion that the sampling process is biased, we clarified the testing procedure in the following manner:

“The TMT is given to all subjects as part of the “core” neuropsychological test battery upon enrollment in the Registry. The Stroop task is given less commonly and is often utilized as a follow-up assessment if impairments in executive function are observed within the core battery. There is no systemic bias in patient selection in terms of lesion location” (Supplementary Note 5).

***Ic.** Very little behavioural analysis is presented, even of the tasks themselves. This all makes it difficult to tell what the authors are really mapping.*

Response: We apologize for not reporting behavioral analysis of TMT and Stroop scores in our sample, the results have been added to the revised manuscript in the following manner:

“In TMT-1, part A scores (mean \pm SD: 42.8 ± 24.0 s) were significantly lower than part B scores (111.4 ± 75.5 s; $t_{334} = -20.82$, $p = 3 \times 10^{-62}$, Cohen’s $d = 1.14$). Similarly, in TMT-2, part A scores (36.8 ± 19.2 s) were significantly lower than part B scores (90.4 ± 49.6 s; $t_{286} = -23.87$, $p = 5 \times 10^{-70}$, Cohen’s $d = 1.41$). In Stroop sample, the score in color-word test (33.8 ± 11.6) is lower than that in color (64.7 ± 16.0) and word tests (87.3 ± 20.7), as reflected in an interference score significantly below zero (defined as $CW - (C+W)/2$; -42.2 ± 12.3 , $t_{228} = -51.89$, $p = 3 \times 10^{-128}$, Cohen’s $d = 3.43$). In sum, in all samples, we observed strong evidence showing slower and/or worse performance in tests requiring more cognitive control (e.g., TMT part B and the color-word Stroop condition), thus supporting the notion that our behavioral measures capture cognitive control.” (Supplementary Note 2)

2. A large number of lesions are analysed but nothing is said about the underlying clinical diagnosis. Were patients with trauma included? The lesion distribution shows a strong frontal preponderance. This reinforces the concern that the sample may be biased.

Response: In the revised manuscript, we added detailed information about the clinical diagnosis in the following table. In particular, patients with head trauma consist only a small portion of each of the three samples (TMT-1: 4%, TMT-2: 3%, Stroop: 5%).

Etiology	TMT-1	TMT-2	Stroop
Stroke / Ischemic	182	154	102
Stroke / Hemorrhagic	66	41	31
Subarachnoid hemorrhage with intraparenchymal lesion	12	14	16
Tumor resection	40	43	45
Resection of arteriovenous malformation, cavernoma, cyst	10	13	13
Resection, Abscess	1	1	1
Trauma with focal intraparenchymal lesion	13	10	11
Encephalitis, herpes simplex	8	5	5
Encephalitis, limbic or other	2	1	1
Other	1	5	4
Total	335	287	229

Supplementary Table 1. Summary of Etiology in each of the three samples.

To further avoid confusion regarding the sampling bias. We provided more details on the sampling procedure in the following manner:

“The sampling procedure involves screening individuals with focal, acquired intraparenchymal lesions that are otherwise neurological and psychiatrically healthy. Many individuals are identified when they are referred for neuropsychological testing. As such, it is possible that this cohort is more enriched in individuals with some degree of cognitive impairment that prompted the referral. In addition, we only enroll patients that are able to participate in cognitive testing, which eliminates individuals with certain types of lesions that result in coma or more severe disability. There is no bias in sampling with regard to the specific hypotheses tested here.”
(Supplementary Note 5)

3. The hypothesis that disconnection of MDN predicts Stroop & TMT performance is primarily tested by comparing the performance of disconnection models including the MDN vs a random selection of a comparable number of ROIs. But given that predictive performance is poor (see Figure 4) a comparison of models is only weak evidence for spatial specificity. If lesion volume or lesions analysed without disconnection produce comparable performance, the argument for specificity becomes hard to sustain. These null models are not provided.

Response: The comparison of prediction performance between CPM on MDN and lesion volume is reported in the manuscript in the following manner:

“CPM using MDN also outperformed prediction using total lesion volume in all three analyses (cross-validation within TMT-1: $\Delta AIC = -13.1$, $p = 0.0014$, $R = 1.02$; TMT-1 \rightarrow TMT-2: $\Delta AIC = -14.2$, $p = 0.0008$, $R = 1.03$; TMT-1 \rightarrow Stroop: $\Delta AIC = -34.6$, $p = 3 \times 10^{-8}$, $R = 1.08$; Figure 4C).” (Page 10)

We also followed the review’s instructions and compared CPM on MDN to predictions based on lesion location without disconnection information (conducted using multivariate lesion symptom

mapping with LESYMAP) and found that CPM outperformed lesion location. The related text reads:

“..., we compared CPM using MDN to predictions based on lesion location without disconnection information (conducted using multivariate lesion symptom mapping with LESYMAP). ... we computed a LESYMAP based on the TMT-1 sample and used it to predict behavioral scores in the TMT-2 and the Stroop samples separately. Both tests showed better performance for CPM using MDN (TMT-1 → TMT-2: $\Delta AIC = -7.8$, $p = 0.020$, $R = 1.01$; TMT-1 → Stroop: $\Delta AIC = -25.2$, $p = 3 \times 10^{-6}$, $R = 1.06$; Figure 4G).” (Page 10)

Figure 4G. Prediction performance using CPM on MDN, compared to that of a lesion-symptom map from TMT-1 (LM) and CPM with cortical ROIs overlapping the lesion-symptom map (CPM+LM).

An additional control model has also been added to the revised manuscript that uses CPM on ROIs selected on account of their overlap with the lesion-symptom map. Please see our response to the comment below for more information.

4. It is not clear why the authors do not compare the lesion-symptom map against the MDN map: the degree of overlap is a direct indicator of the anatomical dependence of the function they are studying. Does a model of the Lesymapped regions, either of lesions or disconnections, predict scores better than MDN out of sample? If it does, then the network that best predicts TMT and

Stroop performance is not the MDN but another set of regions, even if MDN regions might be part of it.

Response: Based on our understanding of the reviewer's suggestion, we conducted additional model comparison in the following manner. We compared out-of-sample predictions from the CPM with MDN ROIs (i.e., our original findings) to the lesion-symptom map. Our results show that the CPM explains more variance than that lesion-symptom map.

Another way that we interpreted this suggestion was to create a new CPM that replaced the MDN ROIs with ROIs selected based on the lesion-symptom map. That is, cortical ROIs that intersected with the lesion-symptom map were selected for a new CPM. Again, the original MDN CPM performed best. This new section is described here:

"..., we performed the CPM on the collection of HCP cortical ROIs that overlapped the LESYMAP above (23% of all ROIs). This CPM numerically outperformed the CPM on MDN in the analysis of TMT-1 → TMT-2 ($\Delta\text{AIC} = -4.5$, $p > 0.095$, $R = 1.01$, Figure 4G). However, it showed statistically significant inferior predictive performance to the CPM on MDN when predicting Stroop performance using information from the TMT-1 sample ($\Delta\text{AIC} = -15.3$, $p = 5 \times 10^{-4}$, $R = 1.03$; Figure 4G)." (Page 10).

In the case that we misunderstood this comment, we will be happy to conduct additional analyses with directions from the reviewer.

5. The minimal degree of overlap between TMT and Stroop casts doubt on the claim that the two tests are measuring the same cognitive function, within the present experimental setup.

Response: We appreciate the concern. First, we should clarify that we are not suggesting that TMT and Stroop measure the exact same thing. Rather, we believe the both tests involve domain-general cognitive control processes. On the other hand, they also involve unique cognitive control processes. As a result, the lesion-symptom maps from both tests should overlap, but not fully overlap.

This point prompted a new analysis evaluating whether the degree of overlap between these two tests was indeed minimal and may be due to chance. Specifically, we generated a distribution of overlap that would be expected from chance and compared our actual overlap to this distribution. This is described here:

“[is associated with impairments in both TMT and Stroop performance (Figure 2B, in green)]. To test if the amount of overlap between the TMT and Stroop lesion-symptom map is significantly above chance, we constructed a null distribution by randomly permuting the locations of clusters that survived multiple comparison for each lesion-symptom map within the brain mask and computed the amount of overlap over 1,000 iterations. The observed overlap is significantly larger than the median of the null distribution ($p = 0.005$), suggesting that the shared area is unlikely to be coincidence.” (Page 5)

We also believe that the concern is related to the debate of domain-general vs. domain-specific cognitive control. We added the following discussion to the revised manuscript to address this issue:

“[...predictors capturing domain-general cognitive control and predictors specialized for individual cognitive control types (i.e., domain-specific cognitive control).] In our experimental design, the TMT and the Stroop task share cognitive control processes that inhibit the processing distractors (e.g., non-target letters and digits in the TMT and the word meaning in the color-word condition of the Stroop task). Inhibition is believed to be a key component of domain-general cognitive control⁷. These shared processes may be reflected in the overlap of the lesion-symptom maps (Fig. 2B) and the overlap of cortical ROIs whose AC disconnection scores significantly scaled with TMT and Stroop measures (Fig. 5A). On the other hand, the TMT and the Stroop tasks also rely on different, domain-specific cognitive control processes. For example, the TMT part B involves cognitive flexibility that switches search target between letter and digits, in contrast to the sustained, stable cognitive control on the color-word interference portion of the Stroop task. Cognitive stability and cognitive flexibility are considered different aspects of cognitive control⁸. Additionally, the two tasks have distinct targets of cognitive control: the TMT

engages cognitive control to modulate visual search, whereas the Stroop task requires cognitive control to modulate stimulus-response associations⁹. Indeed, the co-existence of brain regions supporting domain-general and domain-specific cognitive control is a common finding in the literature^{10,11} and is typically documented as partially overlapping networks, each of which represents cognitive control in a specific domain. We speculated the partially overlapping sets of brain areas for the TMT and the Stroop measures in Fig. 2B and Fig. 5A may capture the co-existence of domain-general and domain-specific cognitive control.” (Page 15-16)

Reviewer #2

0. This is a very important study that should make a valuable contribution to the literature. The Iowa patient cohort is one the best characterized patient cohorts in the world and provides an opportunity to answer questions about brain-behavior relationships that cannot be answered by other available patient cohorts. This study is well-designed, the methods are sound and innovative, the results are clearly presented and the conclusions drawn are supported by the results. However, there are a number of issues that I believe should be addressed that could potentially strengthen the manuscript. These are listed below in no particular order of importance.

Response: We appreciate the reviewer's positive feedback. We describe below the additional analyses and added text that address the reviewer's comments.

1. The etiology of the lesions should be described. The mechanisms of recovery/reorganization of function for tumor resections, traumatic brain injury, and stroke, likely differ. Although it is not clear that this issue affected the results of the study, the authors should acknowledge the heterogeneity of their patients.

Response: We added a new table summarizing the etiology of the lesions in each of the three samples.

Etiology	TMT-1	TMT-2	Stroop
Stroke / Ischemic	182	154	102
Stroke / Hemorrhagic	66	41	31
Subarachnoid hemorrhage with intraparenchymal lesion	12	14	16
Tumor resection	40	43	45
Resection of arteriovenous malformation, cavernoma, cyst	10	13	13
Resection, Abscess	1	1	1
Trauma with focal intraparenchymal lesion	13	10	11
Encephalitis, herpes simplex	8	5	5

Encephalitis, limbic or other	2	1	1
Other	1	5	4
Total	335	287	229

Supplementary Table 1. Summary of Etiology in each of the three samples.

We also added to the revised manuscript the following discussion on the heterogeneity of the etiology:

“Note that we included subjects with different types of lesions (Supplementary Table 1) to increase the sample size and generalizability of the findings. Different etiologies of acquired brain lesions are each associated with specific limitations, such as ischemic stroke lesions being limited to vascular distributions. Inclusion of multiple different lesion etiologies is one way to overcome the limitations of over-reliance on any one etiology. However, one limitation of this approach is that different types of lesions may have different mechanisms of recovery and/or plasticity, which may in turn impede the prediction performance of the model. We acknowledge there is no perfect solution, but we are of the opinion that higher sample sizes with diverse etiologies is preferred relative to smaller samples with a single etiology. When more data is available, future research is encouraged to investigate the similarity of and difference between cognitive control recovery following different types of lesions.” (Supplementary Note 5)

2. The length of time elapsed from the onset of injury to testing should be added to Table 1. Although it is not clear that this issue affected the results of the study, the authors should acknowledge this potential confound.

Response: We added the information to Table 1. The updated Table 1 now reads:

Sample	Group 1 (TMT-1)	Group 2 (TMT-2)	Group 3 (Stroop)
N	335	287	229
Age (SD)	54.54(15.51)	51.71(14.38)	51.09(11.37)
Sex	149(F)/186(M)	139(F)/148(M)	117(F)/112(M)
Education, years (SD)	13.46(2.82)	13.83(2.69)	13.86(2.69)
Handedness	300(R)/26(L)/9(A)	256(R)/24(L)/7(A)	207(R)/17(L)/5(A)

Median (SD) time from onset to test, in months	13.3(63.2)	10.1(91.9)	18.5(98.7)
Lesion laterality	115(R)/157(L)/63(B)	110(R)/110(L)/67(B)	85(R)/83(L)/61(B)
Median lesion size (SD), in cm ³	17.0(43.9)	21.5(81.0)	23.9(78.8)

Table 1. Demographic information for each of the three samples. Abbreviations in Handedness: R = right-handed, L = left-handed, A = ambidextrous.

We did not find significant correlation between the time from lesion onset to assessment and cognitive control performance in any of the three samples. The findings are reported in the revised manuscript in the following manner:

“The duration between lesion onset and behavioral testing was not correlated with cognitive control performance (see below) in any of the three samples (TMT-1: $r = -0.007$, $p > 0.89$; TMT-2: $r = 0.054$, $p > 0.36$; Stroop: $r = -0.079$, $p > 0.23$), indicating that it is unlikely to be a confounding factor in predicting cognitive control performance.” (Page 4)

3. There is a missed opportunity having not investigated other well-documented intrinsic frontal networks derived from resting fMRI data such as the frontoparietal network (FPN), dorsal attention network, ventral attention network, or cingulo-opercular network, all of which have frontal and parietal nodes. As the authors know, the MDN largely overlaps with the FPN network, but recently it has been suggested may also comprise the cingulo-opercular network. At the very least, the authors should make a stronger case in the introduction for only investigating the MDN, as well as note that other frontal networks should be investigated in the future. And in the introduction (line 75), the authors should not refer to the MDN, as “the cognitive control network” since it is one of many involved in cognitive control.

Response: Following the reviewer’s suggestion, we conducted additional analysis comparing prediction performance of the CPM using anatomical connectivity disconnection scores on the MDN, the FPN, the cingulo-opercular network and the dorsal attention network. Please note that we did not include the ventral multimodal network from the HCP atlas, as it only includes four cortical areas. The analysis is reported in the revised manuscript in the following manner:

“Comparison to other networks of cognitive control. A limitation of the current study is that the MDN is only one out of many possible iterations of cognitive control networks. As an exploratory analysis, we compared CPM performance on the MDN to CPM performance on each of three networks that are involved in cognitive control and contribute to regions within the MDN, namely the frontoparietal network, the cingulo-opercular network and the dorsal attention network (Table 3). After FDR correction, the dorsal attention network showed better performance than the MDN in the TMT-1 → TMT-2 analysis ($\Delta AIC = -7.2$, $p = 0.026$, $R = 1.01$). However, MDN outperformed the dorsal attention network in the TMT-1 → Stroop analysis ($\Delta AIC = -10.4$, $p = 0.0055$, $R = 1.02$). No other networks were significantly better than MDN in predicting out-of-sample performance after FDR correction. The overall similar prediction performance may partly attribute to the overlap of the networks and the MDN. More research is needed to further delineate the neuroanatomy of the cognitive control network.” (Page 13)

Network	Analysis	AIC (95% CI)	ΔAIC to MDN	p-value (uncorrected)	R
Frontoparietal network	Within TMT-1	-81.6 (-84.0, -76.9)	1.6	0.312	1.00
	TMT-1 → TMT-2	-153.4 (-219.1, -84.0)	3.6	0.140	1.00
	TMT-1 → Stroop	117.7 (70.4, 154.8)	6.4	0.040	1.01
Cingulo-opercular network	Within TMT-1	-78.5 (-81.5, -73.4)	4.6	0.090	1.01
	TMT-1 → TMT-2	-155.9 (-220.2, -86.7)	1.2	0.358	1.00
	TMT-1 → Stroop	105.2 (56.2, 146.1)	-6.2	0.044	1.01
Dorsal attention network	Within TMT-1	-86.0 (-88.6, -81.5)	-2.9	0.188	1.00
	TMT-1 → TMT-2	-164.3 (-231.1, -93.2)	-7.2	0.026	1.01
	TMT-1 → Stroop	121.7 (77.4, 156.6)	10.4	0.0055	1.02

Table 3. AC-based CPM performance comparison to the MDN. Bold font indicates p-values survived FDR correction. A negative change in AIC reflects better performance and higher positive values reflect worse performance.

We did not intend to give the impression that we regard the MDN as the only definition of the cognitive control network. We do not endorse this view. Rather, it is one instantiation of the

cognitive control network that is accessible to all and convenient to use. In the revised manuscript, we clarified that there are multiple networks that are involved in cognitive control in the following manner:

“Out of the many networks involved in cognitive control, we used the MDN⁸ as an operationalization of the cognitive control network.” (Page 3)

4. The authors make this statement in the results section: “Importantly, the left dorsolateral prefrontal cortex and underlying WM is a region that, when lesioned, is associated with impairments in both TMT and Stroop performance”. Given how precise the anatomical data is in this study, the authors should be precise in their claims. What studies are the authors referring to that have demonstrated that a lesion in a portion of PFC (which looks like the posterior inferior frontal gyrus) causes deficits in the TMT and Stroop. All too often, studies make reference to the lateral PFC in a non-specific way, and this type of statement potentially permeates that type of impreciseness of discussion of lesion-behavior relationships.

Response: Following the reviewer’s comment, we now provide more detailed description of the neuroanatomy of the overlap region. The revised text reads:

“Importantly, the supero-posterior aspect of the left inferior frontal gyrus (area IFSp according to the HCP atlas²⁵) and the underlying WM of the frontal aslant tract is a region included in both the TMT and Stroop lesion-symptom maps and the latent variable derived from both assessments. As such, this region, when lesioned, is associated with greater impairments in both TMT and Stroop performance (Figure 2B, in green, center MNI coordinates = -44, 17, 22).” (Page 5)

5. In general, I prefer that there is little speculation in I discussion sections. However, since this paper likely will open the eyes of many cognitive neuroscientists regarding the importance of white matter damage, the authors have an opportunity to inform the reader regarding why AC is more predictive of neuropsychological deficits than FC, and why white matter damage can be more predictive than cortical damage.

Response: Following the reviewer’s advice, we added the following discussion to the revised manuscript. We also agree with the reviewer on limiting speculations, thus we made the discussion brief.

“First, our results are consistent with other recent work demonstrating better prediction performance using AC-based than FC-based disconnection scores (Figure 4A) and lesion-symptom maps (Figure 4G).¹²⁻¹⁴, 28 Our findings extend this observation to the field of cognitive control. One possible explanation is that GM lesions are better compensated by plasticity mechanisms relative to WM lesions^{15,16}, and thus the location is less predictive of behavioral outcomes long after the lesion.” (Page 15)

Reviewer #3

0. I would like to thank the editor for the opportunity of reviewing this paper on the anatomical and functional disconnection associated with cognitive control deficits. The authors investigated the cognitive control abilities of a large dataset of stroke patients via two classic cognitive control tasks, the TMT and the Stroop task. To specifically explore the role of disconnection in the patients' performance, the patients' anatomical and functional disconnection profiles have been extracted and exploited into prediction models that predicted the performance of an independent sample. Interestingly, the authors implemented a cross-task validation together with a within-group and across-group validation. The results show high performance-prediction accuracy for anatomical connectivity compared to functional connectivity, in particular, left hemisphere connectivity. Moreover, the authors further compare the prediction performance of connectivity of the MDN with whole-brain connectivity demonstrating the role of white matter structures associated with cognitive control.

The study is of potential interest to the readers of Nature Communications. However, a few methodological concerns need to be addressed.

Response: We thank the reviewer for the positive feedback. In our responses, we conducted new analyses and added clarifications to address the comments on the methods.

1. The sample size of the study is impressive. Nevertheless, crucial information on the patients is missing. Could the authors provide more detail on the neuropsychological and neurological status of the patients? For instance, the presence of co-occurring symptoms such as aphasia, neglect, fatigue, hemiplegia and slowness of reaction time could affect the performance in the tasks and hinder the association of the lesion profile to the deficit in cognitive control. Other information, such as the intervals between lesions and assessment/imaging and the lesion size, are missing and would be of potential use in the analysis.

Response: Following the reviewer's comment, we correlated cognitive control performance with neuropsychological test scores of co-occurring symptoms to test whether co-occurring symptoms account for variance in our behavioral measures. The results of these analyses are reported in

response to point 4, below. We also examined the diagnosis notes to identify co-occurring lesion syndromes. The most common symptom is aphasia. However, in this sample the neuropsychological assessments were performed more systematically than and thus provide the optimal method of covarying for these non-target symptoms.

Symptom	TMT-1	TMT-2	Stroop
Aphasia	18	9	3
Neglect	3	0	1
Epilepsy	2	4	0

We also added summary statistics of lesion size and the intervals between lesion onset and test to Table 1 (also cited below).

Sample	Group 1 (TMT-1)	Group 2 (TMT-2)	Group 3 (Stroop)
N	335	287	229
Age (SD)	54.54(15.51)	51.71(14.38)	51.09(11.37)
Sex	149(F)/186(M)	139(F)/148(M)	117(F)/112(M)
Education, years (SD)	13.46(2.82)	13.83(2.69)	13.86(2.69)
Handedness	300(R)/26(L)/9(A)	256(R)/24(L)/7(A)	207(R)/17(L)/5(A)
Median (SD) time from onset to test, in months	13.3(63.2)	10.1(91.9)	18.5(98.7)
Lesion laterality	115(R)/157(L)/63(B)	110(R)/110(L)/67(B)	85(R)/83(L)/61(B)
Median lesion size (SD), in cm ³	17.0(43.9)	21.5(81.0)	23.9(78.8)

Table 1. Demographic information for each of the three samples. Abbreviations in Handedness: R = right-handed, L = left-handed, A = ambidextrous.

We conducted additional control analyses on lesion size and the intervals between lesion onset and test. Please see our response to comment 3 and 4 for more details.

2. What is the rationale for the overlap of the lesions onto the white matter structures associated with the MDM? What information does this analysis provide? Particularly when considering the more informative analyses conducted and described in the following paragraphs.

Response: We thank the reviewer for the opportunity to clarify this point. In the revised manuscript, we now clarify that LESYMAP uses lesion locations to predict behavioral performance and is different from the CPM, which is connectivity-based data. The purpose of the overlap is to test whether lesion locations that can predict cognitive control performance occur within the MDN. We also added an analysis to quantitatively test the prediction. The related changes read:

“[The individual maps were combined using principal component analysis (PCA) to identify common regions of WM connectivity of the MDN ($n = 209$, Figure 2D)]. We tested whether lesion locations that are associated with impaired cognitive control (based on their inclusion in the significant multivariate lesion-symptom maps) are located in the WM tracks linking the MDN. This approach provides a complementary approach for evaluating the anatomy most critical for cognitive control in relation to the MDN anatomy. Specifically, we tested whether a lesion-symptom map overlaps the MDN AC map at above chance level by comparing the size of the overlap to a null distribution of random overlaps. The null distribution was constructed by randomly permuting the locations of the clusters in the lesion-symptom map within a WM mask for 1,000 times. After each permutation, the size of the overlap of the permuted clusters and the MDN AC map was computed as the summation of the voxel-wise dot product of connectivity strength in the MDN AC map and the clusters’ lesion load in the lesion-symptom map. The overlap sizes were pooled as an estimate of the null distribution to derive the p value of the statistical test.

Using the approach above, we found that the conjunction of the TMT and the Stroop lesion-symptom maps (Figure 2B) significantly overlaps with the WM tracks linking the MDN ($p = 0.03$). The overlap is driven by the TMT lesion-symptom map ($p = 0.03$) compared to the Stroop lesion-symptom-map ($p = 0.26$), possibly due to the larger sample size in the former. The lesion-symptom map using the latent variable (Figure 2D) did not significantly overlap with the WM tracks linking the MDN ($p = 0.19$). Despite the Stroop and latent variable maps not reaching significance both intersected with the MDN WM map in the frontal lobe.” (Supplementary Note 4)

Figure 2D. Lesion-symptom map results using a latent variable of cognitive control. Color encodes voxel-level loading in predicting symptom.

3. I am puzzled by how the AC maps are calculated, as they might carry a bias intrinsic to the patient population. Specifically, the score refers to the number of patients who have a disconnection in that specific voxel, disregarding the lesion size and location. This might affect the following analyses. In fact, the successful prediction of the performance of another dataset could be explained by the lesion size and the most likely localisation of the stroke rather than only the goodness of the models. Why did the authors not consider using a more objective score (e.g. the number of streamlines of the healthy participants passing by the specific voxel)? This would allineate the anatomical score with the FC maps score derived from normative data.

Response: We apologize for not being clear when describing our methods. The AC maps are indeed normative maps generated by counting streamlines from an independent sample of healthy subjects. We now clarify the method in the following manner:

“To this end, each cortical ROI from the Human Connectome Project (HCP) atlas¹⁷ was used as a seed region to generate an AC and a functional connectivity (FC) map using tractography and resting-state FC data. These maps were derived from an independent dataset of healthy adults. Next, each lesion mask was overlaid with the AC and FC maps derived from each ROI to produce an AC-based disconnection score (Figure 3A) and an FC-based one.” (Page 7)

And

“*AC disconnection scores.* We used the DSI studio / LEAD-DBS Connectome pipeline (<https://www.lead-dbs.org/about/lead-connectome/>¹⁸) to explore the WM tracts associated with each of the HCP atlas ROIs. As performed previously¹⁹, this software uses normative diffusion-weighted MRI from 32 subjects from the HCP dataset to perform deterministic fiber tractography and estimate the course of WM streamlines associated with a seed region-of-interest²⁰ (<https://ida.loni.usc.edu/>). This data was selected as the state-of-the-art sequences and specialized hardware make it one of the best openly available, in-vivo diffusion MRI datasets available²¹. Diffusion-weighting with $b = 1000 \text{ s/mm}^2$ and 3000 s/mm^2 was applied along 64 directions. Furthermore, additional shells at b -values of 5000 and $10,000 \text{ s/mm}^2$ were applied along 128-directions. In-plane resolution was $1.5 \times 1.5 \text{ mm}$ and 96 slices with a 1.5 mm thickness were acquired. The normative voxel-wise whole brain maps containing the streamlines from each seed ROI were used to generate the AC disconnection scores. This was defined as the voxel-wise sum of each ROI’s WM connectivity map masked by each patient’s lesion map. As the AC maps were generated using an independent dataset, they are not confounded by the lesions in the subjects.” (Page 19)

We agree with the reviewer that lesion volume and locations may also predict behavioral outcomes. To address this issue, we ran control analysis comparing predictive performance between CPMs using lesion volume, lesion locations (i.e., LESYMAP) and AC disconnection scores. The related text reads:

“... CPM using MDN also outperformed prediction using total lesion volume in all three analyses (cross-validation within TMT-1: $\Delta\text{AIC} = -13.1$, $p = 0.0014$, $R = 1.02$; TMT-1 \rightarrow TMT-2: $\Delta\text{AIC} = -14.2$, $p = 0.0008$, $R = 1.03$; TMT-1 \rightarrow Stroop: $\Delta\text{AIC} = -34.6$, $p = 3 \times 10^{-8}$, $R = 1.08$; Figure 4C).

Lastly, we compared CPM using MDN to predictions based on lesion location without disconnection information (conducted using multivariate lesion symptom mapping with lesion-symptom map). Two lesion-symptom map-based predictions were tested. First, we computed a lesion-symptom map-based on the TMT-1 sample and used it to predict behavioral scores in the TMT-2 and the Stroop samples separately. Both tests showed better performance for CPM using MDN (TMT-1 \rightarrow TMT-2: $\Delta\text{AIC} = -7.8$, $p = 0.020$, $R = 1.01$; TMT-1 \rightarrow Stroop: $\Delta\text{AIC} = -25.2$, p

= 3×10^{-6} , $R = 1.06$; Figure 4G). Second, we performed the CPM on the collection of HCP cortical ROIs that overlapped the LESYMAP above (23% of all ROIs). This CPM numerically outperformed the CPM on MDN in the analysis of TMT-1 \rightarrow TMT-2 ($\Delta AIC = -4.5$, $p > 0.095$, $R = 1.01$, Figure 4G). However, it showed statistically significant inferior predictive performance to the CPM on MDN when predicting Stroop performance using information from the TMT-1 sample ($\Delta AIC = -15.3$, $p = 5 \times 10^{-4}$, $R = 1.03$; Figure 4G). Together the results show that AC disconnection in the MDN predicted cross-task cognitive control performance better than a CPM that utilized results of lesion-symptom mapping.” (Page 10)

4. Other variables like lesion size, interval and co-occurring symptoms were not considered in the models. Could the authors comment on their choice and how they prevented these confounding factors from driving their predictions?

Response: We conducted control analysis on lesion size (reported in response to the previous comment) and interval. The added text reporting the results of interval reads:

“The duration between lesion onset and behavioral testing was not correlated with cognitive control performance (see below) in any of the three samples (TMT-1: $r = -0.007$, $p > 0.89$; TMT-2: $r = 0.054$, $p > 0.36$; Stroop: $r = -0.079$, $p > 0.23$), indicating that it is unlikely to be a confounding factor in predicting cognitive control performance.” (Page 4)

To examine whether prediction performance is confounded by co-occurring symptoms, we now report the following analyses:

“To address the possibility of co-occurring symptoms confounding prediction performance, we tested correlations (Spearman’s ρ was used to account for non-normal distributions of some test scores) between neuropsychological scores reflecting five potential co-occurring symptoms and behavioral scores of cognitive control (TMT and Stroop). First, the aphasia severity rating scores from the Boston Diagnostic Aphasia Examination did not correlate with TMT ($n = 135$, $\rho = 0.014$, $p > 0.91$) or Stroop scores ($n = 64$, $\rho = 0.029$, $p > 0.81$). Second, visual neglect ratings did not correlate with TMT ($n = 171$, $\rho = -0.081$, $p > 0.54$) or Stroop scores ($n = 58$, $\rho = -0.10$, $p >$

0.45). Third, to test the confound of hemiplegia, we used grooved pegboard test scores on left and right hand separately. The left minus right contrast was used as a measure of asymmetry in psychomotor performance and was not correlated with TMT ($n = 294$, $\rho = -0.047$, $p > 0.66$) or Stroop scores ($n = 89$, $\rho = 0.073$, $p > 0.49$). The absolute value of this contrast was also employed as a measure of hemiplegia on either side of the body. This measure was not correlated with TMT ($n = 294$, $\rho = 0.007$, $p > 0.94$) or Stroop scores ($n = 89$, $\rho = -0.14$, $p > 0.18$). Lastly, the average of left- and right-hand grooved pegboard test scores was used as a measure of slowing in processing and potential fatigue and did not correlate with TMT ($n = 294$, $\rho = -0.051$, $p > 0.63$) or Stroop scores ($n = 89$, $\rho = -0.063$, $p > 0.55$). For completeness, we also correlated left- and right-hand grooved pegboard test scores separately with TMT (left: $n = 307$, $\rho = -0.094$, $p > 0.36$; right: $n = 299$, $\rho = -0.064$, $p > 0.54$) and Stroop (left: $n = 93$, $\rho = -0.068$, $p > 0.51$; right: $n = 91$, $\rho = -0.098$, $p > 0.35$) scores and did not observe significant correlations. Taken together, we did not find evidence that the cognitive control performance scores were explained by these common co-occurring symptoms.” (Supplementary Note 3)

References

- 1 Egner, T. *The Wiley handbook of cognitive control*. (Wiley Blackwell, 2017).
- 2 Miller, E. K. & Cohen, J. D. An integrative theory of prefrontal cortex function. *Annual review of neuroscience* **24**, 167-202 (2001).
- 3 Bowie, C. R. & Harvey, P. D. Administration and interpretation of the Trail Making Test. *Nature protocols* **1**, 2277-2281 (2006).
- 4 Sánchez-Cubillo, I. *et al.* Construct validity of the Trail Making Test: role of task-switching, working memory, inhibition/interference control, and visuomotor abilities. *Journal of the International Neuropsychological Society* **15**, 438-450 (2009).
- 5 Scarpina, F. & Tagini, S. The stroop color and word test. *Frontiers in psychology* **8**, 557 (2017).
- 6 Mitrushina, M., Boone, K. B., Razani, J. & D'Elia, L. F. *Handbook of normative data for neuropsychological assessment*. (Oxford University Press, 2005).
- 7 Friedman, N. P. & Miyake, A. Unity and diversity of executive functions: Individual differences as a window on cognitive structure. *Cortex; a journal devoted to the study of the nervous system and behavior* **86**, 186-204 (2017).
- 8 Musslick, S. & Cohen, J. D. Rationalizing constraints on the capacity for cognitive control. *Trends in cognitive sciences* (2021).
- 9 Botvinick, M., Braver, T. S., Barch, D. M., Carter, C. S. & Cohen, J. D. Conflict monitoring and cognitive control. *Psychological review* **108**, 624-652 (2001).

- 10 Jiang, J. & Egner, T. Using neural pattern classifiers to quantify the modularity of conflict-control mechanisms in the human brain. *Cerebral cortex* **24**, 1793-1805, doi:10.1093/cercor/bht029 (2014).
- 11 Li, Q. *et al.* Conflict detection and resolution rely on a combination of common and distinct cognitive control networks. *Neuroscience & Biobehavioral Reviews* **83**, 123-131 (2017).
- 12 Salvalaggio, A., De Filippo De Grazia, M., Zorzi, M., Thiebaut de Schotten, M. & Corbetta, M. Post-stroke deficit prediction from lesion and indirect structural and functional disconnection. *Brain : a journal of neurology* **143**, 2173-2188 (2020).
- 13 Griffis, J. C., Metcalf, N. V., Corbetta, M. & Shulman, G. L. Structural disconnections explain brain network dysfunction after stroke. *Cell reports* **28**, 2527-2540. e2529 (2019).
- 14 Reber, J. *et al.* Cognitive impairment after focal brain lesions is better predicted by damage to structural than functional network hubs. *Proceedings of the National Academy of Sciences* **118** (2021).
- 15 Zatorre, R. J., Fields, R. D. & Johansen-Berg, H. Plasticity in gray and white: neuroimaging changes in brain structure during learning. *Nature neuroscience* **15**, 528-536 (2012).
- 16 Sampaio-Baptista, C. & Johansen-Berg, H. White matter plasticity in the adult brain. *Neuron* **96**, 1239-1251 (2017).
- 17 Glasser, M. F. *et al.* A multi-modal parcellation of human cerebral cortex. *Nature* **536**, 171-178, doi:10.1038/nature18933 (2016).
- 18 Horn, A., Ostwald, D., Reisert, M. & Blankenburg, F. The structural–functional connectome and the default mode network of the human brain. *NeuroImage* **102**, 142-151 (2014).
- 19 Horn, A. *et al.* Connectivity predicts deep brain stimulation outcome in Parkinson disease. *Annals of neurology* **82**, 67-78 (2017).
- 20 Setsompop, K. *et al.* Pushing the limits of in vivo diffusion MRI for the Human Connectome Project. *NeuroImage* **80**, 220-233 (2013).
- 21 Wang, Q. *et al.* Normative vs. patient-specific brain connectivity in deep brain stimulation. *NeuroImage* **224**, 117307 (2021).

Reviewer #1 (Remarks to the Author):

The authors have addressed a number of the issues raised, but it remains puzzling that we are not shown a spatial comparison of the disconnectomes elicited by the lesymap analysis and the MDN. The former represents formal inference to the substrates of the behaviour, and if predictive models employing lesymap-derived disconnectomes perform worse than MDN-derived ones, then there is something wrong with the lesion-mapping process. If MDN provides the best explanation then this is what lesymap ought to infer as the lesion-deficit map. If the predictive model based on the lesymap CPM overfits to TMT then it is insufficiently regularized, or TMT and Stroop have meaningfully different substrates which MDN unhelpfully conflates. These issues seem to me to require some discussion.

Reviewer #2 (Remarks to the Author):

The authors have addressed all of my comments in my review.

Reviewer #3 (Remarks to the Author):

I would like to commend the authors for the additional analyses and the clarifications integrated into the original manuscript. As a result, the paper has improved and I recommend it for publication in Nature Communications.

We thank the reviewers for their positive feedback on our last round of revisions. We have further revised the manuscript based on the reviewers' feedback. Please find below our point-by-point responses to the reviewers' comments.

Reviewer #1

- 1. The authors have addressed a number of the issues raised, but it remains puzzling that we are not shown a spatial comparison of the disconnectomes elicited by the lesymap analysis and the MDN. The former represents formal inference to the substrates of the behaviour, and if predictive models employing lesymap-derived disconnectomes perform worse than MDN-derived ones, then there is something wrong with the lesion-mapping process. If MDN provides the best explanation then this is what lesymap ought to infer as the lesion-deficit map. If the predictive model based on the lesymap CPM overfits to TMT then it is insufficiently regularized, or TMT and Stroop have meaningfully different substrates which MDN unhelpfully conflates. These issues seem to me to require some discussion.*

Response: We addressed this comment with several revisions. We review each point in order of appearance, copying the Reviewer comment that we are addressing.

The authors have addressed a number of the issues raised, but it remains puzzling that we are not shown a spatial comparison of the disconnectomes elicited by the lesymap analysis and the MDN.

Following the review's comment, we added a spatial comparison of the anatomical connectivity patterns between the lesymap analysis and the MDN. The added figure is cited below:

Supplementary Figure 3. AC patterns of the MDN (top) and LM (middle) and the difference in AC patterns between MDN and LM (bottom). For each network (MDN or LM), the AC maps seeding from each of the ROIs are averaged and color-coded. Color encodes average streamline numbers from tractography (see Methods). Compared to the LM, MDN AC maps show more anterior connectivity and less connectivity linking inferior parietal and temporal areas. AC: anatomical connectivity; MDN: multiple demand network; LM: lesion-symptom mapping.

The former (referring to lesymap) represents formal inference to the substrates of the behaviour, and if predictive models employing lesymap-derived disconnectomes perform

worse than MDN-derived ones, then there is something wrong with the lesion-mapping process.

We argue that there are two possible reasons that lead to worse performance in lesymap-based results than MDN-based CPM. First, in practice lesymap may have limitations that are not present in the MDN-derived CPM. The lesymap findings are inherently limited to regions of the brain with sufficient lesion coverage to derive significant structure-function relationships. It is possible that with the current sample size and lesion distribution we are only able to identify a portion of the anatomically critical regions. Notably these regions overlap with the MDN. In this scenario it would be reasonable to expect the MDN to outperform the lesymap derived results, which is what we observed. Second, as lesymap is a data-driven approach, it will pick up any information that has most predictive power in the training data. In our case, lesymap may overfit to neural correlates that are domain specific to the TMT, as can be seen in the numerically better predictive performance using lesymap-based CPM than MDN-based CPM (Figure 4g) in the TMT-1 → TMT-2 analysis. However, the overfitting may compromise performance in cross-task analysis (Figure 4g).

If MDN provides the best explanation then this is what lesymap ought to infer as the lesion-deficit map.

Given that there was overlap between the lesymap result and the MDN, our results are consistent with the MDN providing the best explanation of the structure-function relationship. However, additional investigation is needed to determine the relationship of MDN and lesymap results and different aspects of cognitive control.

If the predictive model based on the lesymap CPM overfits to TMT then it is insufficiently regularized, or TMT and Stroop have meaningfully different substrates which MDN unhelpfully conflates. These issues seem to me to require some discussion.

There are different interpretations of the observation. It is possible that performing lesion symptom mapping of TMT includes a mix of neuroanatomical correlates for domain general cognitive control and TMT-specific functions. As the lesymap-based CPM shows similar

performance to MDN-based CPM in within-task analysis (TMT1 → TMT-2 analysis, Figure 4g), we argue that the regularization during training was sufficient for lesymap-based CPM, such that its performance generalizes to a new sample. Instead, what limits the performance of lesymap-based CPM in TMT1 → Stroop analysis is likely to be the inclusion of TMT-specific neural correlates. Given the current design, we are unable to fully test whether the MDN supports non-cognitive control processes shared by the TMT and the Stroop task (but see some control analyses in Supplementary Note 1). More tasks are needed for this purpose.

Overall, while we cannot fully resolve these possibilities in the current analysis, we took this Reviewer's suggestion to add discussion relevant to the finding that the MDN-based connectome-based predictive model (CPM) outperformed CPM based on lesymap-derived CPM in the following manner:

“..., when lesion disconnection scores from all ROIs were used, WM tracts that are only sensitive to one task (e.g., supporting domain-specific cognitive control, Figure 3c and 3d) may be selected in the predictive model, leading to overfitting and worse prediction performance than MDN in the cross-task analysis (Figure 5c). This may also explain the finding that AC-based CPM defined on lesion-symptom map results performed worse than MDN AC-based CPM (Figure 4g) when predicting Stroop task performance using performance in TMT-1, as lesion-symptom mapping method may have relied on neural correlates specific for TMT to predict performance in the Stroop task. To fully investigate the neural substrates of domain-general and domain-specific cognitive control, more tasks capturing cognitive control and more data to further increase lesion coverage are needed.” (Page 10)

Reviewer #2

1. The authors have addressed all of my comments in my review.

Response: We appreciate the reviewer's positive feedback.

Reviewer #3

I. I would like to commend the authors for the additional analyses and the clarifications integrated into the original manuscript. As a result, the paper has improved and I recommend it for publication in Nature Communications.

Response: We thank the reviewer for their kind words.